# WHY VARIANCE REDUCTION HURTS NOISY ZEROTH-ORDER HARD-THRESHOLDING?

## ABSTRACT

The hard-thresholding gradient descent approach is a primary method for solving $\ell_0$-constrained optimization problems to achieve sparsity. In the black-box setting, where only function outputs are accessible, recent work has introduced stochastic and variance-reduced zeroth-order hard-thresholding algorithms to establish both algorithmic and theoretical feasibility, specifically addressing the inherent conflict between zeroth-order and hard-thresholding. However, in practice, function outputs often contain noise, which exacerbates their conflict and undermines the robustness of these algorithms' guarantees. In this work, we investigate the performance for noisy zeroth-order hard-thresholding algorithms, providing convergence analysis for its stochastic version. Furthermore, we theoretically demonstrate the zeroth-order hard-thresholding variance reduction algorithms leveraging historical gradients inherently lowers the tolerable noise upper bound. Contrary to usual presumptions, our findings reveal that variance reduction techniques fail to enhance performance in this setting and even lead to worse feasibility compared to simpler methods without such techniques. These theoretical insights are validated through experiments on a sparse regression problem, black-box adversarial attacks, and biological gene expression.

## 1 INTRODUCTION

The $\ell_0$ constraint plays a pivotal role in machine learning by promoting sparsity, making it a powerful tool for tackling high-dimensional problems. This approach not only mitigates the risk of overfitting, thus enabling consistent and reliable statistical estimation Negahban et al. (2009); Bühlmann & Van De Geer (2011); Raskutti et al. (2011), but also enhances the interpretability and learnability of models Yuan & Li (2021). In this paper, we consider a typical scenario in machine learning where the objective function is decomposed over samples, which leads to the following $\ell_0$-constrained optimization problem:

$$\min_{x \in \mathbb{R}^d} \mathcal{F}(x) = \frac{1}{n} \sum_{i=1}^{n} f_i(x), \quad \text{s.t. } \|x\|_0 \leq k. \tag{1}$$

where $\mathcal{F}(x) : \mathbb{R}^d \to \mathbb{R}$ is a smooth and non-strongly convex loss function, and $\|x\|_0$ denotes the number of nonzero entries in $x$. This problem presents substantial challenges arising from the non-convexity and discrete nature of the $\ell_0$ constraint, which render the corresponding optimization tasks particularly difficult to handle in practice.

When the $\ell_0$ constraint is imposed, the resulting optimization problem becomes NP-hard. A widely adopted technique for addressing such problems is the hard-thresholding gradient descent. Jain et al. (2014) first introduced the full gradient form of hard-thresholding in the Iterative Hard-Thresholding (IHT) algorithm. In the $t$-th iteration, IHT updates the solution as: $x^{t+1} = \mathcal{H}_k(x^t - \eta \nabla \mathcal{F}(x^t))$. Where the hard-thresholding operator $\mathcal{H}_k$ retains the largest $k$ entries while setting the rest to zero. Furthermore, Yuan et al. (2018) and Shen & Li (2018) have proven that the result $\mathcal{H}_k(x)$ is the best $k$-sparse approximation to $x$ in terms of any $\ell_p$ norm $(p \geq 1)$. And unlike other operators, such as the soft-thresholding operator, the hard-thresholding operator has been shown to exhibit expansivity properties Li et al. (2016).

Specifically, Shen & Li (2018) and Li et al. (2016) individually established a bound for the hard-thresholding operator. Building on these developments, Nguyen et al. (2017) extended IHT to its

stochastic gradient descent variant, termed StoIHT, introducing stochasticity to improve scalability and reduce the reliance on full-batch gradient computations. Meanwhile, Li et al. (2016) incorporated a variance reduction technique into the framework to propose SVR-GHT, further enhancing the convergence rate and improving the performance in solving real-world problems. Collectively, these methods leverage first-order gradients to efficiently navigate the optimization landscape while adhering to sparsity constraints.

However, in many real-world scenarios, computing the true gradient is challenging, particularly in black-box problems where only input-output relationships are observable Nesterov & Spokoiny (2017). To address these challenges, zeroth-order (ZO) methods have emerged as a promising alternative, enabling gradient-free optimization through finite difference approximations Brent (2013). These methods have proven to be effective in tackling black-box optimization problems Nesterov & Spokoiny (2017). Furthermore, many ZO algorithms have recently been developed and analyzed, including ZO-SVRG Liu et al. (2018), ZSCG Balasubramanian & Ghadimi (2018).

In particular, de Vazelhes et al. (2022) pioneered the integration of ZO optimization with hard-thresholding, culminating in the development of the Stochastic Zeroth-Order Hard-Thresholding (SZOHT) algorithm. This algorithm was explicitly designed to handle sparsity constraints in gradient-free optimization. However, their theoretical analysis identified a critical limitation related to the number of random directions $q$ used in the estimation of the ZO gradient, resulting from an inherent trade-off between the error of the estimation of the ZO gradient and the expansivity of the hard-thresholding operator. This limitation considerably hinders the practical applicability of the algorithm. To overcome this challenge, Yuan et al. (2024) introduced the SVRG version (VR-SZHT) and $p$-Memorization Stochastic Zeroth-order Hard-Thresholding ($p$M-SZHT) algorithms, leveraging variance reduction techniques. This novel approach alleviates the constraints on ZO gradient steps and mitigates the aforementioned conflict, thereby enhancing convergence rates and expanding the algorithm's range of applications.

Although the above zeroth-order hard-thresholding methods usually work well, their effectiveness in noisy settings of problem equation 1 still remains largely unexplored. Noise frequently arises in practical scenarios, such as stochastic function evaluations or adversarial perturbations. Risteski & Li (2016) has analyzed the convergence accuracy, the information-theoretic lower bound, and the algorithmic upper bound of generalized noisy zeroth-order convex optimization. Then Beznosikov et al. (2020) mixed the zeroth-order stochastic oracle and the first-order oracle to solve the noisy composite optimization problem. Gasnikov et al. (2024) generalize these results to the case where a zeroth-order oracle returns a function value at a point with some adversarial noise. However, whether these results remain consistent under the $\ell_0$ constraint is an open question that requires further investigation. In such settings, the interplay between noise and the zeroth-order hard-thresholding algorithms poses unique challenges, demanding a deeper theoretical and empirical investigation.

In this work, we investigate the performance for noisy zeroth-order hard-thresholding algorithms, including stochastic and variance reduction variants. According to problem equation 1 under noise consideration, our objective function for each component is defined as

$$f_i(x) := \mathbb{E}\left[f_{\xi,i}(x)\right], \quad \text{s.t. } \|x\|_0 \leq k.$$

First, we provide the convergence and complexity analysis of SZOHT relying solely on noise bound condition Risteski & Li (2016); Beznosikov et al. (2020) under the standard assumptions of sparse learning, which are restricted strong smoothness (RSS), and restricted strong convexity (RSC) Nguyen et al. (2017); Shen & Li (2018). Then we combine the variance reduction technique with this setting. We propose a new gradient expression to cover zeroth-order stochastic and some common variance reduction algorithms, theoretically demonstrating that any unbiased zeroth-order hard-thresholding variance reduction algorithms with neighbor gradients inherently lowers the tolerable noise upper bound. Contrary to common results, our findings reveal that variance reduction techniques fail to enhance performance in this setting and even lead to worse feasibility compared to simpler methods without such techniques. These theoretical insights are validated through experiments on a sparse regression problem, black-box adversarial attack, and biological gene expression.

Our contributions can be summarized as follows:

- We extend the stochastic zeroth-order hard-thresholding algorithm to noisy environments and provide a convergence analysis under standard assumptions, along with a necessary condition for validity.

- We also incorporate variance reduction techniques, including the commonly used algorithms, and provide theoretical analysis of their zeroth-order hard-thresholding variants.
- Our theory reveals that variance reduction hurts the performance of noisy zeroth-order hard-thresholding algorithms, and these theoretical results are validated through multiple experiments.

## 2 PRELIMINARIES

### 2.1 ZEROTH-ORDER ESTIMATOR

As described in Liu et al. (2020), de Vazelhes et al. (2022) and Yuan et al. (2024), our zeroth-order gradient estimator is described below:

$$\hat{\nabla} f_{\xi,i}(x) = \frac{d}{q} \sum_{j=1}^{q} \frac{f_{\xi,i}(x + \mu u_j) - f_{\xi,i}(x)}{\mu} u_j. \tag{2}$$

where each random direction $u_j$ is a $d$-dimensional unit vector sampled uniformly from the set $\{u \in R^d : \|u\|_0 \le s_2, \|u\| = 1\}$, $q$ is the number of random unit vectors $u_j$, and $\mu$ is a small constant.

If $s_2 = d$, it turns to the randomized vector-wise gradient estimation like Duchi et al. (2015) and Nesterov & Spokoiny (2017). If $1 \le s_2 < d$, it turns to the generalized coordinate-wise gradient estimation like Lian et al. (2016).

### 2.2 HARD-THRESHOLDING OPERATOR

The hard-thresholding operator $\mathcal{H}_k(\cdot)$ is a non-linear function that preserves the $k$ largest-magnitude components of a vector $x \in \mathbb{R}^n$ while setting all other components to zero.

$$(\mathcal{H}_k(\mathbf{x}))_i = \begin{cases} x_i & |x_i| \text{ is among the } k\text{-th largest values in } x \\ 0 & \text{otherwise} \end{cases}$$

As a simple parameter $\ell_0$ sparsity operation for taking top-$k$, hard-thresholding shows its expansivity. That is, for any $k$-sparse vector $a$ and an arbitrary $d$-dimension vector $b$, we have:

$$\|\mathcal{H}_k(b) - a\| \le \lambda \|b - a\|, \quad \lambda > 1. \tag{3}$$

In earlier studies, a broadly used bound in the literature is $\lambda = 2$. Then Shen & Li (2018) and Li et al. (2016) narrowed down this boundary to $\lambda < 2$, but $\lambda$ still not less than 1. Due to this property, the above inequality means that if $a$ is a feasible solution to the sparse problem, $b$ may not necessarily be closer to $a$ after the hard-thresholding operation. Although iterative gradient hard-thresholding can recover $\|x^t - x^*\| \le \lambda' \|x^{t-1} - x^*\| + Err(x^*)$ satisfying $\lambda' < 1$ Yuan et al. (2018), it doesn't always hold in zeroth-order gradient estimation de Vazelhes et al. (2022).

### 2.3 ASSUMPTIONS

**Assumption 1.** *(Noise bound)Risteski & Li (2016); Beznosikov et al. (2020); Lobanov et al. (2023); Gasnikov et al. (2024) For any noisy function $f_{\xi,i}(x) = f_i(x) + \xi(x)$, there exists a constant $\Delta \ge 0$ such that for any $x$*

$$|f_{\xi,i}(x) - f_i(x)| = |\xi(x)| \le \Delta. \tag{4}$$

*This assumption primarily presupposes a deterministic relationship between the noise-affected function $f_\xi$ and the true function $f$, whereas in reality, the observed discrepancy can stem from multiple sources including sampling errors, stochastic variations in the output, inherent noise within training datasets, and even deliberate adversarial perturbations.*

**Assumption 2.** *(Restricted Strongly Convex, $(m_s, s)$-RSC) Nguyen et al. (2017); Shen & Li (2018); de Vazelhes et al. (2022); Yuan et al. (2024) A differentiable function $\mathcal{F}$ is $m_s$-restricted strongly convex at sparsity level $s$ if there exists a generic constant $m_s > 0$ such that for any $x, y \in R^d$ with $\|x - y\|_0 \le s$,we have:*

$$\mathcal{F}(y) \ge \mathcal{F}(x) + \langle \nabla \mathcal{F}(x), y - x \rangle + \frac{m_s}{2} \|y - x\|^2. \tag{5}$$

**Assumption 3.** *(Restricted Strongly Smooth, $(L_s, s)$-RSS) Nguyen et al. (2017); Shen & Li (2018); de Vazelhes et al. (2022); Yuan et al. (2024) For any $i \in [n]$, $f_i$ is restricted $L_s$-strongly smooth at sparsity level $s$ if there exists a generic constant $L_s > 0$ such that for any $x, y \in R^d$ with $\|x - y\|_0 \leq s$, we have:*

$$f_i(y) \leq f_i(x) + \langle \nabla f_i(x), y - x \rangle + \frac{L_s}{2}\|y - x\|^2. \tag{6}$$

*This assumption implies the following formula to the follows by exchanging $x$ and $y$:*

$$\|\nabla f_i(x) - \nabla f_i(y)\| \leq L_s\|x - y\|. \tag{7}$$

## 3 SZOHT

In this section, we will introduce Stochastic Zeroth-Order Hard-Thresholding (SZOHT) algorithm and give the error bounds of the gradient estimator under noise consideration. Based on the error bounds of the gradient estimator under noise consideration, we provide the convergence and complexity analysis of SZOHT relying solely on noise bound condition under the standard assumptions of sparse learning, which are restricted strongly smooth (RSS), and restricted strongly convex (RSC).

### 3.1 ALGORITHM

To solve black-box sparsity problem, de Vazelhes et al. (2022) proposed SZOHT algorithm, the first zeroth-order sparsity constrained algorithm without assuming any gradient sparsity. They pointed out the conflict between zeroth-order estimation and the expansivity of hard-thresholding operator. This conflict may be exacerbated by the inaccuracy of gradient estimation caused by noise. Now we give the full description of SZOHT under noise consideration as Algorithm 1.

---
**Algorithm 1** Stochastic Zeroth-Order Hard-Thresholding (SZOHT) under noise consideration
---
**Initialization:** Learning rate $\eta$, maximum number of iterations $T$, number of random directions $q$, initial point $x^0$, sparsity level $k$.
**Output:** $x^T$.
**for** $r = 1, ..., T$ **do**
    Randomly sample $i$;
    Compute $\hat{\nabla} f_{\xi,i}(x^{r-1})$;
    $\tilde{x}^r = x^{r-1} - \eta\hat{\nabla} f_{\xi,i}(x^{r-1})$;
    $x^r = \mathcal{H}_k(\tilde{x}^r)$;
**end for**

---

### 3.2 ERROR BOUNDS OF GRADIENT ESTIMATOR UNDER NOISE CONSIDERATION

Now we give the error bounds on the zeroth-order gradient estimator under noise consideration that will be useful in the convergence analysis proof.

**Proposition 1.** *For any support $F \in [d]$ with $|F| = s$, assuming that each $f_i$ satisfies $(L_s, s)$-RSS, for the ZO estimator in , with $q$ random directions, and random supports of size $s_2$. For $\hat{\nabla}_F f_{\xi,i}(x)$, we have:*

$$(a) \ \ \|\mathbb{E}_u\hat{\nabla}_F f_{\xi,i}(x) - \nabla_F f_i(x)\|^2 \leq (L_s\mu + 2\frac{\Delta}{\mu})^2 sd,$$

$$(b) \ \ \mathbb{E}_u\|\hat{\nabla}_F f_{\xi,i}(x)\|^2 \leq (1 + \Delta)(\varepsilon_F\|\nabla_F f_i(x)\|^2 + \varepsilon_{F^c}\|\nabla_{F^c} f_i(x)\|^2 + \varepsilon_{abs}\mu^2)$$

$$+ 4(1 + \frac{1}{\Delta})sd\frac{\Delta^2}{\mu^2}.$$

$$with \ \ \ \varepsilon_{F^c} = \frac{2sd(s_2 - 1)}{q(s_2 + 2)(d - 1)}, \ \ \ \varepsilon_F = \frac{2d}{q(s_2 + 2)}(\frac{(s - 1)(s_2 - 1)}{d - 1} + 3) + 2,$$

$$\varepsilon_{abs} = \frac{2dL_s^2 ss_2}{q}(\frac{(s - 1)(s_2 - 1)}{d - 1} + 1) + L_s^2 sd. \tag{8}$$

From the above inequality, we can see that the noise increases the bias of the gradient estimation, which makes the convergence condition of SZOHT more stringent. From the following analysis, it can be seen that the convergence of SZOHT requires a higher value of $q$ under noise consideration.

*Remark* 1. In typical hard-thresholding algorithms, we always focus on limited dimensions support set $F$, because it is meaningless to focus on other dimensions that are always set to 0.

### 3.3 Convergence analysis

Then we give a convergence analysis result below.

**Theorem 1.** *Suppose $\|x^*\|_0 \leq k^*$, $\mathcal{F}(x)$ satisfies $(m_s, s)$-RSC, the functions $\{f_i(x)\}_{i=1}^n$ satisfies $(L_s, s)$-RSS with $s = 2k + k^*$, and the noisy functions $\{f_{\xi,i}(x)\}_{i=1}^n$ satisfy Assumption 1. When $\eta = \frac{m_s}{(4(1+\Delta)\varepsilon_F+1)L^2}$ and $\frac{d-k^*}{2} \geq k \geq \rho^2 k^*/(1-\rho^2)^2$. Run SZOHT at $r$-th iteration, we have:*

$$\mathbb{E}_u \|x^r - x^*\| \leq (\lambda\rho)^r \|x^0 - x^*\| + \frac{\lambda a}{1-\lambda\rho}\delta + \frac{\lambda b}{1-\lambda\rho}\mu + \frac{\lambda c}{1-\lambda\rho} \cdot \frac{\Delta}{\mu}. \qquad (9)$$

*where $\delta = \|\nabla_F \mathcal{F}(x^*)\|_\infty$, $\rho^2 = 1 - \frac{m_s^2}{(4(1+\Delta)\varepsilon_F+1)L^2}$, $\lambda$ represents the inherent expansion coefficent to the hard-thresholding, and $a = \mathcal{O}(\sqrt{\frac{\Delta}{q}})$, $b = \mathcal{O}(\sqrt{\frac{\Delta}{q}})$, $c = \mathcal{O}(\frac{1}{\sqrt{\Delta}})$ denote the coefficients of different system errors.*

**Corollary 1.** *(Some necessary condition on $q$).* *Assume $\frac{d-k^*}{2} \geq k \geq \rho^2 k^*/(1-\rho^2)^2$, in order to ensure that $\lambda\rho < 1$, we need the following necessary (but not sufficient) condition on $q$:*

*Case $s_2 > 1$:*

$$q_{min} = \frac{16(1+\Delta)d\kappa^2(s_2-1)k^*}{(s_2+2)(d-1)}[2\varepsilon_\Delta - 1 + 2\sqrt{\varepsilon_\Delta(\varepsilon_\Delta-1) + \frac{1}{2} - \frac{1}{2k^*} + \frac{3}{2}\frac{d-1}{k^*(s_2-1)}}]. \qquad (10)$$

*Case $s_2 = 1$:*

$$q_{min} = \frac{8(1+\Delta)\kappa^2 d}{\sqrt{\frac{d}{k^*}}+1}. \quad with \quad \kappa = \frac{L}{m_s}, \varepsilon_\Delta = (9+4\Delta)\kappa^2. \qquad (11)$$

*Remark* 2. Theorem 1 and Corollary 1 present convergence results for the parameter and a convergence condition when the algorithm is extended to noisy settings. The former contains a linear convergence term $\lambda\rho$ and system errors from hard-thresholding, zeroth-order gradient estimation, and noise. Significantly, the coefficient $c$ of the last term caused by noise will never vanish, even with a large $q$. The latter reveals the requirement of hard-thresholding for gradient estimation, that is, the larger the noise $\Delta$, the more query times are needed. All of these results are transformed into de Vazelhes et al. (2022)'s if $\Delta = 0$. They show that SZOHT degrades under noise consideration, but is still effective.

## 4 Variance reduction algorithms

Compared to its full gradient counterpart, SZOHT exhibits a relatively low sublinear convergence rate, as it is a variant of SGD, primarily due to the variance in stochastic gradients introduced by random sampling Gu et al. (2020). Additionally, ZO gradient estimates estimates exacerbate this issue by introducing high variance Liu et al. (2018). To address these challenges, various variance reduction techniques have been developed and analyzed in recent years, including SVRG Johnson & Zhang (2013), SAGA Defazio et al. (2014) and $q$-memorization Hofmann et al. (2015). Building on these advancements, Yuan et al. (2024) proposed VR-SZHT and $p$M-SZHT extending SZOHT with variance reduction techniques derived from SVRG and $q$-memorization.

In this section, we use the $\alpha$-unbiased gradient estimation $\hat{g} = \hat{\nabla}f_{\xi,i}(x) - \alpha(\hat{\nabla}f_{\xi,i}(\varphi) - \mathbb{E}\hat{\nabla}f_{\xi,i}(\varphi))$ for iteration (where $\varphi_t$ means historical parameters before $t$-th iteration) to enable a comparison of SZOHT, VR-SZHT, and $p$M-SZHT within the same analytical framework. This ensures that when $\alpha = 0$, the algorithm becomes SZOHT, and when $\alpha = 1$, the algorithm becomes VR-SZHT and $p$M-SZHT. To distinguish them from previous algorithms, we refer to them as $\alpha$-VR-SZHT and $\alpha$-$p$M-SZHT.

## 4.1 $\alpha$-VR-SZHT

$\alpha$-VR-SZHT is the zeroth-order hard-thresholding variant of SVRG, a widely-used variance reduction technique designed to accelerate gradient-based optimization algorithms. Similarly to SVRG, $\alpha$-VR-SZHT employs an outer loop and an inner loop structure. In each outer iteration, the algorithm computes a full gradient to serve as a variance-reducing reference, while each inner iteration executes an iteration involving stochastic updates followed by a hard-thresholding operation to promote sparsity. The complete procedure for $\alpha$-VR-SZHT is detailed in Algorithm 2.

---

**Algorithm 2** $\alpha$-unbiased Variance Reduced Stochastic Zeroth-Order Hard-Thresholding($\alpha$-VR-SZHT) under noise consideration

---

**Initialization:** Learning rate $\eta$, maximum number of iterations $T$, update frequency $m$, number of random directions $q$, initial point $x^0$, sparsity level $k$.
**Output:** $x^T$.
**for** $r = 1, ..., T$ **do**
$\quad x^{(0)} = x^{r-1}$;
$\quad \hat{\mu} = \frac{1}{n} \sum_{i=1}^{n} \hat{\nabla} f_{\xi,i}(x^{(0)})$;
$\quad$ **for** $t = 0, ..., m - 1$ **do**
$\quad\quad$ Randomly sample $i_r \in \{1, \ldots, n\}$;
$\quad\quad$ Compute $\hat{\nabla} f_{\xi,i_r}(x^{(t)}), \hat{\nabla} f_{\xi,i_r}(x^{(0)})$;
$\quad\quad \hat{g}^{(t)} = \hat{\nabla} f_{\xi,i_r}(x^{(t)}) - \alpha(\hat{\nabla} f_{\xi,i_r}(x^{(0)}) - \hat{\mu})$;
$\quad\quad \tilde{x}^{(t)} = x^{(t)} - \eta \hat{g}^{(t)}$;
$\quad\quad x^{(t+1)} = \mathcal{H}_k(\tilde{x}^{(t)})$;
$\quad$ **end for**
$\quad x^r = x^{(m)}$;
**end for**

---

**Theorem 2.** *Suppose $\|x^*\|_0 \leq k^*$, $\mathcal{F}(x)$ satisfies $(m_s, s)$-RSC, the functions $\{f_i(x)\}_{i=1}^n$ satisfies $(L_s, s)$-RSS with $s = 2k + k^*$, and the noisy functions $\{f_{\xi,i}(x)\}_{i=1}^n$ satisfy Assumption 1. Run $\alpha$-VR-SZHT after $m$ iterations, we have:*

$$\gamma[\mathbb{E}\mathcal{F}(x^{(r)}) - \mathcal{F}(x^*)] \leq \rho_\alpha[\mathbb{E}\mathcal{F}(x^{(r-1)}) - \mathcal{F}(x^*)] + \lambda \frac{\beta^m - 1}{\beta - 1}(L_\mu + L_\Delta + L_{\Delta,\mu} + L_\delta)$$

$$+ L_r \mathbb{E}_{u,i_r} \|x^{(r-1)} - x^*\|^2. \tag{12}$$

*where $\gamma = (2\eta - 48(1 + \Delta)\eta^2 \varepsilon_F L_s)\lambda \frac{\beta^m - 1}{\beta - 1}, \rho_\alpha = 2 \frac{\beta^m}{m_s} + 48\alpha^2(1 + \Delta)\eta^2 \lambda \varepsilon_F$, and $\beta$ is given by $(1 + \eta^2 m_s^2)\lambda$. $L_\delta, L_r$ denote the inherent errors, and $L_\mu \sim \mathcal{O}(\frac{\mu^2}{q})$, $L_\Delta \sim \mathcal{O}(\frac{\Delta}{q})$, $L_{\Delta,\mu} \sim \mathcal{O}(\frac{\Delta^2}{\mu^2})$ denote some estimation errors.*

**Corollary 2.** *(**The tolerable upper bound of noise for $\alpha$-VR-SZHT**). According to the above inequality, we give the upper bound of noise that ensures $\rho_\alpha < \gamma$ to achieve convergence of $\alpha$-VR-SZHT as following:*

$$\Delta < \frac{\eta(\beta^m - 1)\lambda m_s - (\beta - 1)\beta^m}{24\eta^2(\beta^m - 1)\lambda m_s \varepsilon_F L_s(\alpha^2 + 1)} - 1. \tag{13}$$

*Remark* 3. Theorem 2 and Corollary 2 present convergence results for the value of the function and the tolerable upper bound $\Delta$ when the algorithm is extended to noisy settings. $\alpha$-VR-SZHT ensures a decrease in the function value within every inner loop of $m$ iterations. Furthermore, the minimum constraint on $q$, as discussed in Corollary 1, can be removed by appropriately controlling the size of $m$. However, while the algorithm reduces variance effectively, noise becomes a limiting factor, particularly when the step size $\eta$ is very small. Corollary 2 demonstrates that the tolerable upper bound for noise is negatively correlated with $\alpha$, highlighting that while variance reduction enhances convergence, it also introduces a trade-off by increasing the sensitivity of the algorithm to noise.

## 4.2   $\alpha$-$p$M-SZHT

Different from $\alpha$-VR-SZHT, $\alpha$-$p$M-SZHT selects a random index set $J \subseteq \{1, .., n\}$ in each iteration, and have the probability $\frac{p}{n}$ to update gradient $\alpha_i$ as

$$\alpha_{i_r}^{(r)} = \begin{cases} \hat{\nabla} f_{\xi, i_r}(x^{(r)}) & if \ \ i_r \in J, \\ \alpha_{i_r}^{(r-1)} & otherwise. \end{cases} \tag{14}$$

where $p$ is the number of directions updated each iteration. As a generalized algorithm framework, $\alpha$-$p$M-SZHT implements variance reduction with neighbors. It contains zeroth-order hardthresholding variants of some common variance reduction algorithms. If $J = \{1, ..., n\}$, it turns into $\alpha$-VR-SZHT with the number of iterations in the inner loop $m = 1$. If $|J| = 1$ and $p = n$, it turns into SAGA-ZOHT. We give full description of $\alpha$-$p$M-SZHT as Algorithm 3.

---

**Algorithm 3** $\alpha$-unbiased Stochastic Variance Reduced Zeroth-Order Hard-Thresholding with $p$-Memorization ($\alpha$-$p$M-SZHT) under noise consideration

---

**Initialization:** Learning rate $\eta$, maximum number of iterations $T$, number of random directions $q$, initial point $x^0$, sparsity level $k$.
**Output:** $x^T$.
**for** $r = 1, ..., T$ **do**
  Update $\hat{a}^{(r-1)}$;
  Randomly sample $i_r \in \{1, \ldots, n\}$;
  $\hat{g} = \hat{\nabla} f_{\xi, i_r}(x^{r-1}) - \alpha(\hat{a}_{i_r}^{(r-1)} - \frac{1}{n} \sum_{i=1}^n \hat{a}_{i_r}^{(r-1)})$;
  $\tilde{x}^r = x^{r-1} - \eta \hat{g}$;
  $x^r = \mathcal{H}_k(\tilde{x}^r)$;
**end for**

---

**Theorem 3.** *Suppose $\|x^*\|_0 \le k^*$, $\mathcal{F}(x)$ satisfies $(m_s, s)$-RSC, the functions $\{f_i(x)\}_{i=1}^n$ satisfies $(L_s, s)$-RSS with $s = 2k + k^*$, and the noisy functions $\{f_{\xi, i}(x)\}_{i=1}^n$ satisfy Assumption 1. Run $\alpha$-$p$M-SZHT at $r$-th iteration, we have:*

$$\mathbb{E}\mathcal{F}(x^{(r)}) - \mathcal{F}(x^*) \le \rho_\alpha'(\mathbb{E}\mathcal{F}(x^{(r-1)}) - \mathcal{F}(x^*)) + L_\delta' + L_{\Delta, \mu}'. \tag{15}$$

*where $\rho_\alpha' = \frac{2\beta}{m_s} + 48\eta^2 \lambda(1 + \Delta)\varepsilon_F L_s - 2\eta\lambda + 2\alpha^2(1 + \Delta)(1 - \frac{p}{n})$ and $L_\delta'$, $L_{\Delta, \mu}'$ denote some different system errors.*

**Corollary 3.** *(**The tolerable upper bound of noise for $\alpha$-$p$M-SZHT**). According to the above inequality, we give the upper bound of noise that ensures $\rho_\alpha' < 1$ to achieve convergence of $\alpha$-$p$M-SZHT as following:*

$$\Delta < \frac{(\frac{1}{2} + \eta\lambda)m_s - \beta}{m_s(24\eta^2 \lambda \varepsilon_F L_s + \alpha^2(1 - \frac{p}{n}))} - 1. \tag{16}$$

*Remark* 4. As an application of another variance reduction framework, $\alpha$-$p$M-SZHT obtains different but similar convergence result for function values in Theorem 3. It contains a set of inevitable inherent errors and a convergence coefficient, which lead to the analogous tolerable upper bound $\Delta$ in Corollary 3.

## 4.3   VARIANCE REDUCTION HURTS NOISY ZEROTH-ORDER HARD-THRESHOLDING

The above conclusions analyze the convergence results from the perspective of specific algorithms, presenting the relationship between the convergence conditions of noisy zeroth-order hardthresholding algorithms and its parameters. It's now reasonable to assert that as the for a given noise level $\Delta$, using historical gradients for variance reduction ($\alpha > 0$) amplifies the effect of the noise, resulting in larger system errors and worse convergence rates.

Now we will discuss this issue from another perspective of variance reduction. Assume there is no noise, many existing works, such as Liu et al. (2018), have proven that those techniques do reduce the variance of zeroth-order gradient estimation as:

$$\|\hat{g} - \nabla\mathcal{F}(x)\|^2 \le \|\hat{\nabla} f_i(x) - \nabla\mathcal{F}(x)\|^2. \tag{17}$$

where $\hat{g} = \hat{\nabla} f_i(x) - \alpha(\hat{\nabla} f_i(\varphi) - \mathbb{E}\hat{\nabla} f_i(\varphi))$. However, in the presence of noise, even when the noise is unbiased, i.e., $\mathbb{E}_\xi[f_{\xi,i}(x)] = f_i(x)$, the following holds:

$$\|\hat{g}_\xi - \nabla\mathcal{F}(x)\|^2 \leq (1+\Delta)\|\hat{\nabla} f_i(x) - \nabla\mathcal{F}(x)\|^2 + 4sd(1 + \frac{1}{\Delta})(1+2\alpha)^2\frac{\Delta^2}{\mu^2}. \qquad (18)$$

This can also lead to the conclusion that very small noise can easily break inequality 17, and more queries caused by variance reduction further amplifies the impact of noise. Therefore, in this setting, those variance reduction algorithms struggle to achieve the goal of reducing variance. Instead, they accumulate greater noise effects, making gradient estimation less accurate, which results in more excessive expansivity of hard-thresholding.

## 5 EXPERIMENTS

In this section, we evaluate the performance of SZOHT, SAGA-ZOHT (as a specific algorithm of $\alpha$-$p$M-SZHT), and VR-SZHT through a series of experiments. The latter two algorithms represent two different variance reduction techniques. We conducted the following three experiments on one simulated dataset and two real-world datasets. For fair comparison, we denote the number of iterations by NHT (number of hard-thresholding operations) rather than IZO (iterative zeroth-order oracle). In fact, using IZO to calculate the number of iterations for SAGA-ZOHT and VR-SZHT will be much more than NHT.

### 5.1 SENSITIVITY ANALYSIS

We begin by conducting experiments on a toy example to validate our conclusion. We consider the following noisy problem: $f_{\xi,i}(x) = (a_i^T x - b_i)^2 + \xi(x)$ with $a \in \mathbb{R}^{n\times d}, x \in \mathbb{R}^d$. The noise $\xi(x)$ is randomly from uniform distribution $\mathcal{U}(-\Delta, \Delta)$ for each output. In our experiments, we consider $n = 100, d = 100, k = 30$ and randomly generate $a$ and $x$ from a uniform distribution and set $b = a^T x$. We fix the parameters $q = 20, \mu = 10^{-4}, s_2 = d = 100, \eta = 10^{-6}$ and $m = 5$ for inner iteration number of VR-SZHT, varying the noise bound $\Delta$ in $\{0, 10, 20, 30, 40\}$ across SZOHT, SAGA-ZOHT and VR-SZHT. Our results are shown in figure 1.

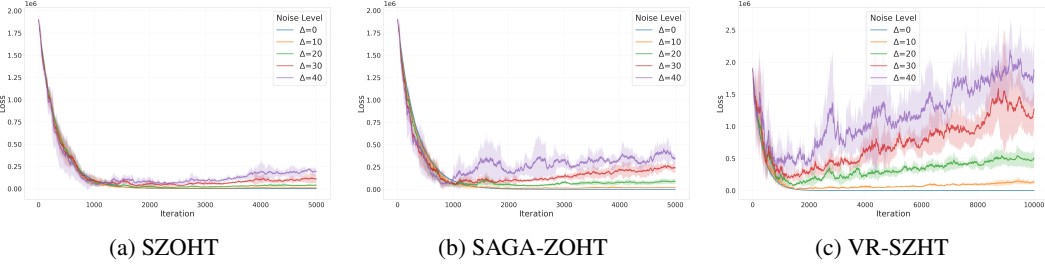

| (a) SZOHT | (b) SAGA-ZOHT | (c) VR-SZHT |

Figure 1: Comparison of convergence performance among SZOHT, SAGA-ZOHT, and VR-SZHT optimization algorithms. The solid line of the curve represents the mean of multiple experiments, while the shaded area represents the variance.

### 5.2 FEW PIXELS ADVERSARIAL ATTACKS AGAINST DEFENSE

Next, we consider the problem of adversarial attacks with few pixels against a lightweight defense method, Random Noise Defense (RND) Qin et al. (2021). The objective is formulated as $\max_\zeta f_\xi(x + \zeta)$ such that $\|\zeta\|_0 \leq k$, where $x$ means images and $f_\xi$ represents cross-entropy loss function with RND defense applied to a pre-trained model. RND introduces a random Gaussian noise for each query, defined as follows: $f_{\xi,i} = f_i(x + \nu v)$ where $v \sim \mathcal{N}(0, I)$ and $\nu$ are selected by the defender. In our experiments, we pre-trained a simple CNN LeCun et al. (1998a) on MNIST LeCun et al. (1998b), then selected 100 images which are correctly classified as the number 7, and set $k = 50, q = 100$ to attack under different $\nu$. Our attack optimization curves are shown in figure 3.

*Remark* 5. For the cross-entropy loss function in CNNs, it is believed that a smaller perturbation expectation, $\nu v$, results in function noise which could be bounded by a smaller constant $\Delta$.

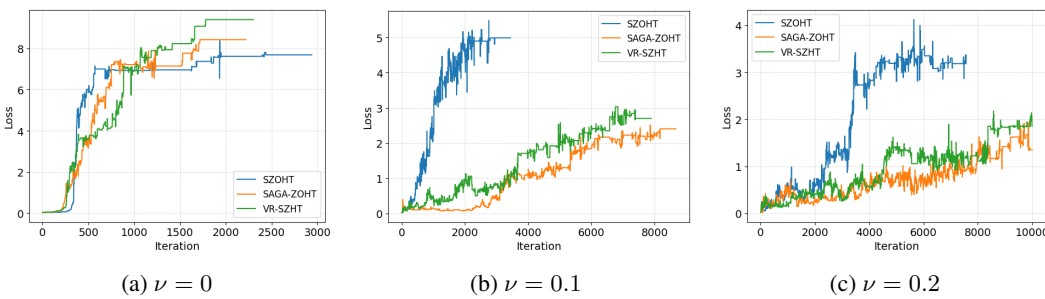

(a) $\nu = 0$        (b) $\nu = 0.1$        (c) $\nu = 0.2$

Figure 3: Increase in attack loss under $\ell_0$-constrained attacks. The three algorithms perform consistently without noise, while SZOHT still performs well in the presence of increased noise.

## 5.3 BIOLOGICAL GENE EXPRESSION

We also conduct a biological gene expression experiment. Gene expression profiling allows for the detection of numerous genes; however, only a small subset of these genes significantly influences the expression of disease traits. Moreover, gene expression data obtained through artificial detection often contain substantial noise. While manual denoising is one approach to address this issue, predicting diseases using noisy gene data remains a significant challenge. In this study, we aim to identify a sparse vector $x \in \mathbb{R}^d$ from gene expression matrix $A \in \mathbb{R}^{n \times d}$ and a predictive model $f$. This sparse vector helps to more accurately capture the genes truly responsible for disease traits, thereby improving the precision of disease prediction. In our experiment, we used a breast cancer dataset[1] comprising 286 samples and 16,382 genes. The dataset was divided into training and testing set. A deep neural network (DNN) was pre-trained on the training set, achieving an accuracy of 66.67% on the testing set. Subsequently, we utilized the training set to $\min_x f(Ax^T)$. Our results are shown in figure 2.

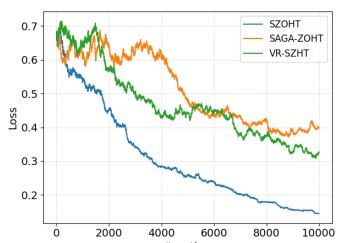

Figure 2: Convergence performance on biological gene expression experiment by comparing three different zeroth-order hard-thresholding algorithms.

## 6 CONCLUSION

In this paper, we analyze the black-box $\ell_0$ sparse optimization problem, and extend the zeroth-order hard-thresholding algorithms to noise scenarios. Our theoretical analysis demonstrates that the convergence performance of SZOHT, while degraded in noisy scenarios, remains effective. Additionally, we specify the parameter condition for the convergence of SZOHT, which is the number of zeroth-order gradient directions $q$. We further investigated the performance of variance reduction techniques and proposed a novel zeroth-order gradient representation to account for whether variance reduction is employed. We analyze two types of variance reduction techniques, the zeroth-order hard-thresholding variants of SVRG and $q$-Memorization, namely $\alpha$-VR-SZHT and $\alpha$-$p$M-SZHT. Our theoretical analysis proved that these variance reduction techniques reduce the tolerable noise upper bound, ultimately leading to inferior results compared to SZOHT, and face significant challenges in effectively reducing variance in this scenario. Our results highlight the challenges and limitations of applying variance reduction to noisy zeroth-order hard-thresholding. These findings are validated through multiple experiments. Our work may inspire advancements in various fields, such as zeroth-order fine-tuning of Large Language Model. It is also an interesting question whether the limitations of these algorithms can be mitigated by employing specific methods to handle noise effectively.

---

[1]https://www.ncbi.nlm.nih.gov/geo/query/acc.cgi?acc=GSE2034

ETHICS STATEMENT

All participants in this work, as well as the paper submission, adhere to the ICLR Code of Ethics ( https://iclr.cc/public/CodeOfEthics).

REPRODUCIBILITY STATEMENT

We affirm that the results of this work are fully reproducible. Appendix provides the theoretical proofs. And the source code will be publicly released after publication of the paper.

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

## A  NOTATIONS AND DEFINITIONS

Throughout this appendix, we will use the following notations:

- $\nabla f(x)$ denotes the gradient of $f$ at $x$.

- $[d]$ denotes the set of all integers between 1 and $d : \{1, .., d\}$.

- $u_i$ denotes the $i$-th coordinate of vector $u$.

- $\| \cdot \|_0$ denotes the $\ell_0$ norm.

- $\| \cdot \|$ denotes the $\ell_2$ norm.

- $\| \cdot \|_\infty$ denotes the maximum absolute component of a vector.

- $\mathrm{supp}(x)$ denotes the support of a vector $x$, that is the set of its non-zero coordinates.

- $|F|$ denotes the cardinality (number of elements) of a set $F$.

- All the sets we consider are subsets of $[d]$. So for a given set $F$, $F^c$ denotes the complement of $F$ in $[d]$.

- $\boldsymbol{u}_F$ (resp. $\nabla f(x)$) denotes the hard-thresholding of $\boldsymbol{u}$ (resp. $\nabla f(x)$) over the support $F$, that is, a vector which keeps $\boldsymbol{u}$ (resp. $\nabla f(x)$) untouched for the set of coordinates in $F$, but sets all other coordinates to 0.

- $\xi(x)$ denotes the value of the random noise at $x$ and $\Delta$ denotes the upper bound of $\xi(x)$.

## B  SYMBOL DEFINITION

We have placed the symbol definitions that were not provided in detail in the main text here.

### B.1  THEOREM 1

$$\lambda^2 = 1 + \frac{k^*/k + \sqrt{(4 + k^*/k)k^*/k}}{2},$$

$$a = (\sqrt{4(1 + \Delta)[\varepsilon_F s + \varepsilon_{F^c}(d - k)] + 2s} + \sqrt{s})\eta,$$

$$b = \frac{L_{s_2}\sqrt{sd}}{L} + \sqrt{2(1 + \Delta)\varepsilon_{abs}}\eta,$$

$$c = 2\frac{\sqrt{sd}}{L} + \sqrt{8(1 + \frac{1}{\Delta})sd}\eta. \tag{19}$$

### B.2 Theorem 2

$$\gamma = (2\eta - 48(1+\Delta)\eta^2 \varepsilon_F L_s)\lambda \frac{\beta^m - 1}{\beta - 1},$$

$$\beta = (1 + \eta^2 m_s^2)\lambda,$$

$$\rho_\alpha = 2\frac{\beta^m}{m_s} + 48\alpha^2(1+\Delta)\eta^2 \lambda \varepsilon_F,$$

$$\delta' = \mathbb{E}_{i_r}\|\nabla f_{i_r}(x^*)\|_\infty^2,$$

$$L_r = 2\frac{\beta^m}{m_s}\sqrt{s}\delta,$$

$$L_\mu = (\frac{sdL_s^2}{m_s^2} + 6\eta^2(1+\alpha^2)(1+\Delta)\varepsilon_{abs})\mu^2,$$

$$L_\Delta = 4\frac{sdL_s}{m_s^2}\Delta + 3\eta^2(1+\alpha^2)(4\varepsilon_F s + 4\varepsilon_{F^c}(d-k))\delta'^2\Delta$$

$$L_{\Delta,\mu} = 24\frac{sd\eta^2\Delta^2}{\mu^2}(1+\alpha^2)(1+\frac{1}{\Delta}) + 4\frac{sd\Delta^2}{\mu^2 m_s^2},$$

$$L_\delta = 3\eta^2 \mathbb{E}_{i_r}\|(1-\alpha)\nabla_F f_{i_r}(x^*) + \alpha\nabla_F \mathcal{F}(x^*)\|^2 + 3\eta^2(1+\alpha^2)((4\varepsilon_F + 2)s + 4\varepsilon_{F^c}(d-k))\delta'^2. \tag{20}$$

### B.3 Theorem 3

$$L_\delta' = 6\eta^2 \lambda s((1-\alpha)^2\delta'^2 + \alpha^2\delta^2),$$

$$L_{\Delta,\mu}' = \eta^2\lambda(6(1+\Delta)\varepsilon_{abs}\mu^2 + 24sd(1+\frac{1}{\Delta})\frac{\Delta^2}{\mu^2} + 3\eta^2\lambda\alpha^2 A_{r-1} + \frac{\lambda}{m_s^2}(\sqrt{\varepsilon_\mu\mu^2} + 2\frac{\Delta}{\mu}\sqrt{sd})^2$$

$$+ 3\eta^2\lambda\tau\delta'^2,$$

$$A_r = \frac{1}{n}\sum_{k=1}^{r-1}\left(1-\frac{p}{n}\right)^{r-k-1}(2(1+\Delta)\varepsilon_{abs}\mu^2 + 8sd(1+\frac{1}{\Delta})\frac{\Delta^2}{\mu^2})$$

$$+ \frac{1}{n}\sum_{k=1}^{r-1}\left(1-\frac{p}{n}\right)^{r-k-1}((4(1+\Delta)\varepsilon_F + 2)s + 4(1+\Delta)\varepsilon_{F^c}(d-k))\mathbb{E}_{i_r}\|\nabla f_i(x^*)\|_\infty^2$$

$$+ \left(1-\frac{p}{n}\right)^r(2(1+\Delta)\varepsilon_{abs}\mu^2 + 8sd(1+\frac{1}{\Delta})\frac{\Delta^2}{\mu^2})$$

$$+ \left(1-\frac{p}{n}\right)^r((4(1+\Delta)\varepsilon_F + 2)s + 4(1+\Delta)\varepsilon_{F^c}(d-k))\mathbb{E}_{i_r}\|\nabla f_i(x^*)\|_\infty^2,$$

$$\tau = (4(1+\Delta)\varepsilon_F + 2)s + 4(1+\Delta)\varepsilon_{F^c}(d-k). \tag{21}$$

## C Proof of SZOHT

### C.1 Proof of Proposition 1

**Lemma 1.** (Proof in de Vazelhes et al. (2022), Lemma B.2) For any support $F \in [d]$ with $|F| = s$, we have:

$$\mathbb{E}_u\|u_F\| \leq \sqrt{sd}, \tag{22}$$

$$\mathbb{E}_u\|u_F\|^2 = \frac{s}{d}, \tag{23}$$

$$\mathbb{E}_u\|u_F\|^4 = \frac{(s+2)s}{(d+2)d}. \tag{24}$$

**Lemma 2.** (Proof in de Vazelhes et al. (2022), Lemma C.3) For any support $F \in [d]$ with $|F| = s$, assuming that each $f_i$ satisfies $(L_s, s)$-RSS, for the ZO estimator described in out work, with $q$

random directions, and random supports of size $s_2$, we have:

$$(a) \quad \|\mathbb{E}_u \hat{\nabla}_F f_i(x) - \nabla_F f_i(x)\|^2 \le \varepsilon_\mu \mu^2,$$

$$(b) \quad \mathbb{E}_u \|\hat{\nabla}_F f_i(x)\|^2 \le \varepsilon_F \|\nabla_F f_i(x)\|^2 + \varepsilon_{F^c} \|\nabla_{F^c} f_i(x)\|^2 + \varepsilon_{abs} \mu^2$$

$$with \quad \varepsilon_\mu = L_s^2 sd, \quad \varepsilon_F = \frac{2d}{q(s_2+2)}(\frac{(s-1)(s_2-1)}{d-1}+3) + 2,$$

$$\varepsilon_{F^c} = \frac{2sd(s_2-1)}{q(s_2+2)(d-1)}, \quad \varepsilon_{abs} = \frac{2dL_s^2 ss_2}{q}(\frac{(s-1)(s_2-1)}{d-1}+1) + L_s^2 sd.$$

$$(25)$$

**Proposition 1.** For any support $F \in [d]$ with $|F| = s$, assuming that each $f_i$ satisfies $(L_s, s)$-RSS, for the ZO estimator in 1, with $q$ random directions, and random supports of size $s_2$. For SZOHT with $\hat{\nabla}_F f_{\xi,i}(x)$, we have:

$$(a) \quad \|\mathbb{E}_u \hat{\nabla}_F f_{\xi,i}(x) - \nabla_F f_i(x)\|^2 \le (L_s \mu + 2\frac{\Delta}{\mu})^2 sd,$$

$$(b) \quad \mathbb{E}_u \|\hat{\nabla}_F f_{\xi,i}(x)\|^2 \le (1+\Delta)(\varepsilon_F \|\nabla_F f_i(x)\|^2 + \varepsilon_{F^c} \|\nabla_{F^c} f_i(x)\|^2 + \varepsilon_{abs} \mu^2) + 4(1+\frac{1}{\Delta})sd\frac{\Delta^2}{\mu^2}$$

$$with \quad \varepsilon_{F^c} = \frac{2sd(s_2-1)}{q(s_2+2)(d-1)}, \varepsilon_F = \frac{2d}{q(s_2+2)}(\frac{(s-1)(s_2-1)}{d-1}+3) + 2,$$

$$\varepsilon_{abs} = \frac{2dL_s^2 ss_2}{q}(\frac{(s-1)(s_2-1)}{d-1}+1) + L_s^2 sd.$$

$$(26)$$

*Proof.* For Proposition 1(a), we denote by by $\hat{\nabla} f_{\xi,i}(x) = \frac{d}{q}\sum_{k=1}^q \frac{f_{\xi,i}(x+\mu u_k)-f_{\xi,i}(x)}{\mu} u_k$ the ZO gradient estimator from 1, we have:

$$\|\mathbb{E}_u \hat{\nabla}_F f_{\xi,i}(x) - \nabla_F f_i(x)\|$$

$$= \|\mathbb{E}_u \frac{1}{q}\sum_{k=1}^q d\frac{f_{\xi,i}(x+\mu u_k)-f_{\xi,i}(x)}{\mu} u_{k,F} - \nabla_F f_i(x)\|$$

$$= \|\frac{d}{q}\mathbb{E}_u \sum_{k=1}^q \frac{f_i(x+\mu u_k)-f_i(x)+\xi(x+\mu u_k)-\xi(x)}{\mu} u_{k,F} - \nabla_F f_i(x)\|$$

$$\le \|d\mathbb{E}_u \frac{f_i(x+\mu u)-f_i(x)}{\mu} u_F - \nabla_F f_i(x)\| + \|d\mathbb{E}_u \frac{\xi(x+\mu u)-\xi(x)}{\mu} u_F\|$$

$$= \|\mathbb{E}_u \hat{\nabla}_F f_i(x) - \nabla_F f_i(x)\| + \|d\mathbb{E}_u \frac{\xi(x+\mu u)-\xi(x)}{\mu} u_F\|$$

$$\overset{(a)}{\le} \sqrt{\varepsilon_\mu \mu^2} + \frac{d}{\mu}\mathbb{E}_u \|\xi(x+\mu u)-\xi(x)\| \cdot \|u_F\|$$

$$\overset{(b)}{\le} \sqrt{\varepsilon_\mu \mu^2} + \frac{d}{\mu}\sqrt{\mathbb{E}_u \|\xi(x+\mu u)-\xi(x)\|^2 \cdot \mathbb{E}_u \|u_F\|^2}$$

$$\le \sqrt{\varepsilon_\mu \mu^2} + 2\frac{\Delta}{\mu}\sqrt{sd}$$

$$(27)$$

Where (a) follows Lemma 2(a) and (b) follows by the Cauchy-Schwarz inequality for random variables: $\mathbb{E}(XY) \le \sqrt{\mathbb{E}(X^2)\mathbb{E}(Y^2)}$. The last inequality follows Lemma 1 and the Assumption 1. For

Proposition 1(b), we have:

$$\mathbb{E}_u\|\hat{\nabla}_F f_{\xi,i}(x)\|^2$$

$$= \mathbb{E}_u\|\frac{d}{q}\sum_{k=1}^{q}\frac{f_{\xi,i}(x+\mu u_k)-f_{\xi,i}(x)}{\mu}u_{k,F}\|^2$$

$$= \mathbb{E}_u\|\frac{d}{q}\sum_{k=1}^{q}\frac{f_i(x+\mu u_k)-f_i(x)+\xi(x+\mu u_k)-\xi(x)}{\mu}u_{k,F}\|^2$$

$$= \mathbb{E}_u\|\hat{\nabla}_F f_i(x)+\frac{d}{q}\sum_{i=1}^{q}\frac{\xi(x+\mu u_k)-\xi(x)}{\mu}u_{k,F}\|^2$$

$$\leq \mathbb{E}_u\|\hat{\nabla}_F f_i(x)\|^2+\frac{d^2}{q^2}\mathbb{E}_u\|\sum_{k=1}^{q}\frac{\xi(x+\mu u_k)-\xi(x)}{\mu}u_{k,F}\|^2$$

$$+2\frac{d}{q}\mathbb{E}_u\|\hat{\nabla}_F f_i(x)\|\cdot\|\sum_{k=1}^{q}\frac{\xi(x+\mu u_k)-\xi(x)}{\mu}u_{k,F}\|$$

$$\overset{(a)}{\leq} (1+\Delta)\mathbb{E}_u\|\hat{\nabla}_F f_i(x)\|^2+\frac{d^2}{q^2}(1+\frac{1}{\Delta})\mathbb{E}_u\|\sum_{k=1}^{q}\frac{\xi(x+\mu u_k)-\xi(x)}{\mu}u_{k,F}\|^2$$

$$\tag{28}$$

Where (a) follows from the inequality $2\langle u,v\rangle \leq m\|u\|^2 + \frac{1}{m}\|v\|^2$. For $\mathbb{E}_u\|\sum_{k=1}^{q}\frac{\xi(x+\mu u_k)-\xi(x)}{\mu}u_{k,F}\|^2$, using Assumption 1, we can get:

$$\mathbb{E}_u\|\sum_{k=1}^{q}\frac{\xi(x+\mu u_k)-\xi(x)}{\mu}u_{k,F}\|^2 \leq 4\frac{\Delta^2}{\mu^2}\mathbb{E}_u(\sum_{k=1}^{q}\|u_{k,F}\|)^2 \tag{29}$$

Taking 29 into 28, we get:

$$(1+\Delta)\mathbb{E}_u\|\hat{\nabla}_F f_i(x)\|^2+\frac{d^2}{q^2}(1+\frac{1}{\Delta})\mathbb{E}_u\|\sum_{k=1}^{q}\frac{\xi(x+\mu u_k)-\xi(x)}{\mu}u_{k,F}\|^2$$

$$\leq (1+\Delta)\mathbb{E}_u\|\hat{\nabla}_F f_i(x)\|^2+4\frac{d^2}{q^2}(1+\frac{1}{\Delta})\frac{\Delta^2}{\mu^2}\mathbb{E}_u(\sum_{k=1}^{q}\|u_{k,F}\|)^2$$

$$= (1+\Delta)\mathbb{E}_u\|\hat{\nabla}_F f_i(x)\|^2+4\frac{d^2}{q^2}(1+\frac{1}{\Delta})\frac{\Delta^2}{\mu^2}\sum_{i=1}^{q}\sum_{j=1}^{q}\mathbb{E}_u\|u_{i,F}\|\|u_{j,F}\|$$

$$\overset{(a)}{\leq} (1+\Delta)\mathbb{E}_u\|\hat{\nabla}_F f_i(x)\|^2+4\frac{d^2}{q^2}(1+\frac{1}{\Delta})\frac{\Delta^2}{\mu^2}\sum_{i=1}^{q}\sum_{j=1}^{q}\sqrt{\mathbb{E}_u\|u_{i,F}\|^2\cdot\mathbb{E}_u\|u_{j,F}\|^2}$$

$$\overset{(b)}{\leq} (1+\Delta)\varepsilon_F\|\nabla_F f_i(x)\|^2+(1+\Delta)\varepsilon_{F^c}\|\nabla_{F^c} f_i(x)\|^2+(1+\Delta)\varepsilon_{abs}\mu^2+4sd(1+\frac{1}{\Delta})\frac{\Delta^2}{\mu^2}$$

$$\tag{30}$$

Where (a) follows from the inequality $\mathbb{E}(XY)\leq\sqrt{\mathbb{E}(X^2)\mathbb{E}(Y^2)}$ and (b) follows Lemma 2(b).

## C.2 PROOF OF THEOREM 1

Before providing the proof of Theorem 1, we need the following lemma:

**Lemma 3** Hard-Thresholding's expansivity de Vazelhes et al. (2022) quelity(27):
For any $k^*$-sparse vector $x^*$, we have:

$$\|\mathcal{H}_k(x)-x^*\|^2 \leq \lambda^2\|x-x^*\|^2, \quad \lambda^2 = 1+\frac{k^*/k+\sqrt{(4+k^*/k)k^*/k}}{2} \tag{31}$$

**Theorem 1** Suppose $\mathcal{F}(x)$ satisfies assumption 2,and the functions $\{f_i(x)\}_{i=1}^n$ satisfy assumption 3 with $s = 2k + k^*$, and the noisy functions $\{f_{\xi,i}(x)\}_{i=1}^n$ satisfy assumption 1. When $\eta = \frac{m_s}{(4(1+\Delta)\varepsilon_F + 1)L^2}$ and $\frac{d-k^*}{2} \geq k \geq \rho^2 k^*/(1 - \rho^2)^2$. Run SZOHT at $r$ iteration,we have:

$$\mathbb{E}\|x^{(r)} - x^*\| \leq (\lambda\rho)^r \|x^0 - x^*\| + \frac{\lambda a}{1 - \lambda\rho}\delta + \frac{\lambda b}{1 - \lambda\rho}\mu + \frac{\lambda c}{1 - \lambda\rho} \cdot \frac{\Delta}{\mu},$$

$$with \quad \rho^2 = 1 - \frac{m_s^2}{(4(1+\Delta)\varepsilon_F + 1)L^2},$$

$$\lambda^2 = 1 + \frac{k^*/k + \sqrt{(4 + k^*/k)k^*/k}}{2},$$

$$a = (\sqrt{(4(1+\Delta)[\varepsilon_F s + \varepsilon_{F^c}(d - k)] + 2s} + \sqrt{s})\eta,$$

$$b = \frac{L_{s_2}\sqrt{sd}}{L} + \sqrt{2(1+\Delta)\varepsilon_{abs}}\eta,$$

$$c = 2\frac{\sqrt{sd}}{L} + \sqrt{8(1 + \frac{1}{\Delta})sd}\eta.$$

$$(32)$$

$Proof.$ First we focus on $\|y^{(r)} - x^*\|$ with $y^{(r)} = x^{(r)} - \eta\hat{\nabla}_F \mathcal{F}_\xi(x^{(r)})$.

$$\mathbb{E}_u\|y^{(r)} - x^*\| = \mathbb{E}_u\|x^{(r)} - \eta\hat{\nabla}_F \mathcal{F}_\xi(x^{(r)}) - x^*\|$$
$$= \mathbb{E}_u\|x^{(r)} - x^* - \eta\hat{\nabla}_F \mathcal{F}_\xi(x^{(r)}) + \eta\nabla_F \mathcal{F}(x^*) - \eta\nabla_F \mathcal{F}(x^*)\|$$
$$\leq \mathbb{E}_u\|x^{(r)} - x^* - \eta\hat{\nabla}_F \mathcal{F}_\xi(x^{(r)}) + \eta\nabla_F \mathcal{F}(x^*)\| + \eta\|\nabla_F \mathcal{F}(x^*)\|$$

$$(33)$$

Then we try to get the upper bound of $\mathbb{E}_u\|x^{(r)} - x^* - \eta\hat{\nabla}_F \mathcal{F}_\xi(x^{(r)}) + \eta\nabla_F \mathcal{F}(x^*)\|$.

$$\mathbb{E}_u\|x^{(r)} - x^* - \eta\hat{\nabla}_F \mathcal{F}_\xi(x^{(r)}) + \eta\nabla_F \mathcal{F}(x^*)\|^2$$
$$= \|x^{(r)} - x^*\|^2 + \eta^2 \mathbb{E}_u\|\hat{\nabla}_F \mathcal{F}_\xi(x^{(r)}) - \nabla_F \mathcal{F}(x^*)\|^2$$
$$\quad - 2\eta\langle x^{(r)} - x^*, \mathbb{E}_u\hat{\nabla}_F \mathcal{F}_\xi(x^{(r)}) - \nabla_F \mathcal{F}(x^*)\rangle$$
$$= \|x^{(r)} - x^*\|^2 + \eta^2 \mathbb{E}_u\|\hat{\nabla}_F \mathcal{F}_\xi(x^{(r)}) - \nabla_F \mathcal{F}(x^*)\|^2$$
$$\quad - 2\eta\langle x^{(r)} - x^*, \mathbb{E}_u\hat{\nabla}_F \mathcal{F}_\xi(x^{(r)}) - \nabla_F \mathcal{F}(x^{(r)})\rangle - 2\eta\langle x^{(r)} - x^*, \nabla_F \mathcal{F}(x^{(r)}) - \nabla_F \mathcal{F}(x^*)\rangle$$
$$\overset{(a)}{\leq} \|x^{(r)} - x^*\|^2 + \eta^2 \mathbb{E}_u\|\hat{\nabla}_F \mathcal{F}_\xi(x^{(r)}) - \nabla_F \mathcal{F}(x^*)\|^2 + \eta^2 L^2\|x^{(r)} - x^*\|^2$$
$$\quad + \frac{1}{L^2}\|\mathbb{E}_u\hat{\nabla}_F \mathcal{F}_\xi(x^{(r)}) - \nabla_F \mathcal{F}(x^{(r)})\|^2 - 2\eta\langle x^{(r)} - x^*, \nabla_F \mathcal{F}(x^{(r)}) - \nabla_F \mathcal{F}(x^*)\rangle$$
$$\overset{(b)}{\leq} \|x^{(r)} - x^*\|^2 + \eta^2 \mathbb{E}_u\|\hat{\nabla}_F \mathcal{F}_\xi(x^{(r)}) - \nabla_F \mathcal{F}(x^*)\|^2 + \eta^2 L^2\|x^{(r)} - x^*\|^2$$
$$\quad + \frac{1}{L^2}\|\mathbb{E}_u\hat{\nabla}_F \mathcal{F}_\xi(x^{(r)}) - \nabla_F \mathcal{F}(x^{(r)})\|^2 - 2\eta m_s\|x^{(r)} - x^*\|^2$$
$$= (1 - 2\eta m_s + \eta^2 L^2)\|x^{(r)} - x^*\|^2 + \eta^2 \mathbb{E}_u\|\hat{\nabla}_F \mathcal{F}_\xi(x^{(r)}) - \nabla_F \mathcal{F}(x^*)\|^2$$
$$\quad + \frac{1}{L^2}\|\mathbb{E}_u\hat{\nabla}_F \mathcal{F}_\xi(x^{(r)}) - \nabla_F \mathcal{F}(x^{(r)})\|^2$$
$$\leq (1 - 2\eta m_s + \eta^2 L^2)\|x^{(r)} - x^*\|^2 + 2\eta^2 \mathbb{E}_u\|\hat{\nabla}_F \mathcal{F}_\xi(x^{(r)})\|^2 + 2\eta^2\|\nabla_F \mathcal{F}(x^*)\|^2$$
$$\quad + \frac{1}{L^2}\|\mathbb{E}_u\hat{\nabla}_F \mathcal{F}_\xi(x^{(r)}) - \nabla_F \mathcal{F}(x^{(r)})\|^2$$

$$(34)$$

Where (a) follows from $2\langle u, v\rangle \leq m\|u\|^2 + \frac{1}{m}\|v\|^2$, (b) follows the definition of $(m_s, s)$-RSC with writing respectively at $x = x^{(r)}, y = x^*$.

Use Proposition 1(a)(b), we have:

$$\mathbb{E}_u \|x^{(r)} - x^* - \eta \hat{\nabla}_F \mathcal{F}_\xi(x^{(r)}) + \eta \nabla_F \mathcal{F}(x^*)\|^2$$

$$\leq (1 - 2\eta m_s + \eta^2 L^2)\|x^{(r)} - x^*\|^2 + \frac{1}{L^2}(L_{s_2}\mu + 2\frac{\Delta}{\mu})^2 sd + 8(1 + \Delta)\eta^2 sd\frac{\Delta^2}{\mu^2}$$

$$+ 2\eta^2\|\nabla_F \mathcal{F}(x^*)\|^2 + 2(1 + \Delta)(\varepsilon_F \eta^2\|\nabla_F \mathcal{F}(x^{(r)})\|^2 + \varepsilon_{F^c}\eta^2\|\nabla_{F^c}\mathcal{F}(x^{(r)})\|^2 + \varepsilon_{abs}\eta^2\mu^2)$$

$$\leq (1 - 2\eta m_s + \eta^2 L^2)\|x^{(r)} - x^*\|^2 + \frac{1}{L^2}(L_{s_2}\mu + 2\frac{\Delta}{\mu})^2 sd + 8(1 + \Delta)\eta^2 sd\frac{\Delta^2}{\mu^2}$$

$$+ 2\eta^2\|\nabla_F \mathcal{F}(x^*)\|^2 + 2(1 + \Delta)\varepsilon_F \eta^2[2\|\nabla_F \mathcal{F}(x^{(r)}) - \nabla_F \mathcal{F}(x^*)\|^2 + 2\|\nabla_F \mathcal{F}(x^*)\|^2]$$

$$+ 2\|\nabla_{F^c}\mathcal{F}(x^*))\|^2] + 2(1 + \Delta)\varepsilon_{abs}\eta^2\mu^2 + 2(1 + \Delta)\varepsilon_{F^c}\eta^2[2\|\nabla_{F^c}\mathcal{F}(x^{(r)}) - \nabla_{F^c}\mathcal{F}(x^*)\|^2$$

$$\leq (1 - 2\eta m_s + \eta^2 L^2)\|x^{(r)} - x^*\|^2 + \frac{1}{L^2}(L_{s_2}\mu + 2\frac{\Delta}{\mu})^2 sd$$

$$+ 4(1 + \Delta)\varepsilon_F \eta^2\|\nabla_F \mathcal{F}(x^{(r)}) - \nabla_F \mathcal{F}(x^*)\|^2 + 4(1 + \Delta)\varepsilon_{F^c}\eta^2\|\nabla_{F^c}\mathcal{F}(x^{(r)}) - \nabla_{F^c}\mathcal{F}(x^*)\|^2$$

$$+ ((4(1 + \Delta)\varepsilon_F + 2)s + 4(1 + \Delta)\varepsilon_{F^c}(d - k))\eta^2\|\nabla_F \mathcal{F}(x^*)\|_\infty^2 + 2(1 + \Delta)\varepsilon_{abs}\eta^2\mu^2$$

$$+ 8(1 + \Delta)\eta^2 sd\frac{\Delta^2}{\mu^2}$$

$$(35)$$

Using the fact $\varepsilon_{F^c} \leq \varepsilon_F$ and the definition of the Euclidean norm, we get:

$$\mathbb{E}_u \|x^{(r)} - x^* - \eta \hat{\nabla}_F \mathcal{F}_\xi(x^{(r)}) + \eta \nabla_F \mathcal{F}(x^*)\|^2$$

$$\leq (1 - 2\eta m_s + \eta^2 L^2)\|x^{(r)} - x^*\|^2 + \frac{1}{L^2}(L_{s_2}\mu + 2\frac{\Delta}{\mu})^2 sd$$

$$+ 4(1 + \Delta)\varepsilon_F \eta^2\|\nabla \mathcal{F}(x^{(r)}) - \nabla \mathcal{F}(x^*)\|^2 + ((4(1 + \Delta)\varepsilon_F + 2)s$$

$$+ 4(1 + \Delta)\varepsilon_{F^c}(d - k))\eta^2\|\nabla_F \mathcal{F}(x^*)\|_\infty^2 + 2(1 + \Delta)\varepsilon_{abs}\eta^2\mu^2 + 8(1 + \Delta)\eta^2 sd\frac{\Delta^2}{\mu^2}$$

$$\overset{(a)}{\leq} (1 - 2\eta m_s + (4(1 + \Delta)\varepsilon_F + 1)\eta^2 L^2)\|x^{(r)} - x^*\|^2 + \frac{1}{L^2}(L_{s_2}\mu + 2\frac{\Delta}{\mu})^2 sd$$

$$+ ((4(1 + \Delta)\varepsilon_F + 2)s + 4(1 + \Delta)\varepsilon_{F^c}(d - k))\eta^2\|\nabla_F \mathcal{F}(x^*)\|_\infty^2 + 2(1 + \Delta)\varepsilon_{abs}\eta^2\mu^2$$

$$+ 8(1 + \Delta)\eta^2 sd\frac{\Delta^2}{\mu^2}$$

$$= (1 - 2\eta m_s + (4(1 + \Delta)\varepsilon_F + 1)\eta^2 L^2)\|x^{(r)} - x^*\|^2 + ((4(1 + \Delta)\varepsilon_F + 2)s$$

$$+ 4(1 + \Delta)\varepsilon_{F^c}(d - k))\eta^2\delta^2 + \frac{1}{L^2}(L_{s_2}\mu + 2\frac{\Delta}{\mu})^2 sd + 2(1 + \Delta)\varepsilon_{abs}\eta^2\mu^2 + 8(1 + \frac{1}{\Delta})\eta^2 sd\frac{\Delta^2}{\mu^2}$$

$$(36)$$

Where (a) follows from $(L, s' := max(s_2, s))$-RSS.

Thus, because of $\sqrt{a+b} \le \sqrt{a} + \sqrt{b}$, we get

$$
\begin{aligned}
\mathbb{E}_u \|x^{(r)} &- x^* - \eta \hat{\nabla}_F \mathcal{F}_\xi(x^{(r)}) + \eta \nabla_F \mathcal{F}(x^*)\| \\
&\le \sqrt{1 - 2\eta m_s + (4(1+\Delta)\varepsilon_F + 1)\eta^2 L^2} \|x^{(r)} - x^*\| \\
&\quad + \sqrt{(4(1+\Delta)\varepsilon_F + 2)s + 4(1+\Delta)\varepsilon_{F^c}(d-k)}\eta\delta \\
&\quad + \frac{1}{L}(L_{s_2}\mu + 2\frac{\Delta}{\mu})\sqrt{sd} + \sqrt{2(1+\Delta)\varepsilon_{abs}}\eta\mu + \sqrt{8(1+\frac{1}{\Delta})sd}\eta\frac{\Delta}{\mu} \\
&\le \sqrt{1 - 2\eta m_s + (4(1+\Delta)\varepsilon_F + 1)\eta^2 L^2} \|x^{(r)} - x^*\| \\
&\quad + \sqrt{(4(1+\Delta)\varepsilon_F + 2)s + 4(1+\Delta)\varepsilon_{F^c}(d-k)}\eta\delta \\
&\quad + (\frac{L_{s_2}\sqrt{sd}}{L} + \sqrt{2(1+\Delta)\varepsilon_{abs}}\eta)\mu + (2\frac{\sqrt{sd}}{L} + \sqrt{8(1+\frac{1}{\Delta})sd}\eta)\frac{\Delta}{\mu}
\end{aligned}
\tag{37}
$$

Plug into $\mathbb{E}_u \|y^{(r)} - x^*\|$:

$$
\begin{aligned}
\mathbb{E}_u \|y^{(r)} &- x^*\| \\
&\le \sqrt{1 - 2\eta m_s + (4(1+\Delta)\varepsilon_F + 1)\eta^2 L^2} \|x^{(r)} - x^*\| \\
&\quad + \sqrt{(4(1+\Delta)\varepsilon_F + 2)s + 4(1+\Delta)\varepsilon_{F^c}(d-k)}\eta\delta \\
&\quad + (\frac{L_{s_2}\sqrt{sd}}{L} + \sqrt{2(1+\Delta)\varepsilon_{abs}}\eta)\mu + (2\frac{\sqrt{sd}}{L} + \sqrt{8(1+\frac{1}{\Delta})sd}\eta)\frac{\Delta}{\mu} + \eta\|\nabla_F \mathcal{F}(x^*)\| \\
&\le \sqrt{1 - 2\eta m_s + (4(1+\Delta)\varepsilon_F + 1)\eta^2 L^2} \|x^{(r)} - x^*\| \\
&\quad + (\sqrt{(4(1+\Delta)\varepsilon_F + 2)s + 4(1+\Delta)\varepsilon_{F^c}(d-k)} + \sqrt{s})\eta\delta \\
&\quad + (\frac{L_{s_2}\sqrt{sd}}{L} + \sqrt{2(1+\Delta)\varepsilon_{abs}}\eta)\mu + (2\frac{\sqrt{sd}}{L} + \sqrt{8(1+\frac{1}{\Delta})sd}\eta)\frac{\Delta}{\mu}
\end{aligned}
\tag{38}
$$

To minimize the left term, we define the following $\rho$ with fixed $\eta = \frac{m_s}{(4(1+\Delta)\varepsilon_F + 1)L^2}$

$$
\rho = \sqrt{1 - \frac{m_s^2}{(4(1+\Delta)\varepsilon_F + 1)L^2}}
\tag{39}
$$

Then we define the following $a, b$ and $c$ for simplicity

$$
\begin{aligned}
a &= (\sqrt{(4(1+\Delta)\varepsilon_F + 2)s + 4(1+\Delta)\varepsilon_{F^c}(d-k)} + \sqrt{s})\eta \\
b &= \frac{L_{s_2}\sqrt{sd}}{L} + \sqrt{2(1+\Delta)\varepsilon_{abs}}\eta \\
c &= 2\frac{\sqrt{sd}}{L} + \sqrt{8(1+\frac{1}{\Delta})sd}\eta
\end{aligned}
\tag{40}
$$

Therefore we have:

$$
\mathbb{E}_u \|y^{(r)} - x^*\| \le \rho\|x^{(r)} - x^*\| + a\delta + b\mu + c\frac{\Delta}{\mu}
\tag{41}
$$

Combine Hard-Thresholding with $\lambda = \sqrt{1 + \frac{k^*/k + \sqrt{(4+k^*/k)k^*/k}}{2}}$

$$
\mathbb{E}_u \|x^{(r+1)} - x^*\| \le \lambda\rho\|x^{(r)} - x^*\| + \lambda a\delta + \lambda b\mu + \lambda c\frac{\Delta}{\mu}
\tag{42}
$$

We need $\lambda\rho \le 1$, similar to de Vazelhes et al. (2022),

$$
k \ge \rho^2 k^*/(1-\rho^2)^2
\tag{43}
$$

Summing from $0$ to $t$, we get:

$$\mathbb{E}\|x^{(r+1)} - x^*\| \le (\lambda\rho)^{r+1}\|x^0 - x^*\| + \sum_{i=0}^{r}(\lambda\rho)^i(\lambda a\delta + \lambda b\mu + \lambda c\frac{\Delta}{\mu})$$

$$= (\lambda\rho)^{r+1}\|x^0 - x^*\| + \frac{1 - (\lambda\rho)^r}{1 - \lambda\rho}(\lambda a\delta + \lambda b\mu + \lambda c\frac{\Delta}{\mu})$$

$$\le (\lambda\rho)^{r+1}\|x^0 - x^*\| + \frac{1}{1 - \lambda\rho}(\lambda a\delta + \lambda b\mu + \lambda c\frac{\Delta}{\mu})$$

$$(44)$$

Where the last inequality follows the fact $\lambda\rho \le 1$.

**Corollary 1.** There is a necessary(but not sufficient)condition on $q$.
Case $s_2 > 1$:

$$q_{min} = \frac{16(1 + \Delta)d\kappa^2(s_2 - 1)k^*}{(s_2 + 2)(d - 1)}[2\varepsilon_\Delta - 1 + 2\sqrt{\varepsilon_\Delta(\varepsilon_\Delta - 1) + \frac{1}{2} - \frac{1}{2k^*} + \frac{3}{2}\frac{d - 1}{k^*(s_2 - 1)}}],$$

$$with \quad \varepsilon_\Delta = (9 + 4\Delta)\kappa^2.$$

$$(45)$$

Case $s_2 = 1$:

$$q_{min} = \frac{8(1 + \Delta)\kappa^2 d}{\sqrt{\frac{d}{k^*}} + 1}.$$

$$(46)$$

*Proof.* To ensure convergence, we need $\lambda\rho \le 1$, it leads to $k \ge \rho^2 k^*/(1 - \rho^2)^2$

$$\lambda^2 = 1 + \frac{k^*/k + \sqrt{(4 + k^*/k)k^*/k}}{2}$$

$$\rho^2 = 1 - \frac{1}{(4(1 + \Delta)\varepsilon_F + 1)\kappa^2}$$

$$with \quad \varepsilon_F = \frac{2d}{q(s_2 + 2)}(\frac{(s - 1)(s_2 - 1)}{d - 1} + 3) + 2, \quad s = 2k + k^*, \quad \kappa = \frac{L}{m_s}$$

$$(47)$$

So we get

$$\rho^2 = 1 - \frac{1}{[\frac{8(1+\Delta)d}{q(s_2+2)}(\frac{(s-1)(s_2-1)}{d-1} + 3) + 9 + 4\Delta]\kappa^2}$$

$$= 1 - \frac{1}{[\frac{8(1+\Delta)d}{q(s_2+2)}(\frac{(2k+k^*-1)(s_2-1)}{d-1} + 3) + 9 + 4\Delta]\kappa^2}$$

$$(48)$$

Define $a := \frac{16(1+\Delta)d\kappa^2(s_2-1)}{q(s_2+2)(d-1)}$ and $b := \kappa^2[\frac{8(1+\Delta)d}{q(s_2+2)}(\frac{(k^*-1)(s_2-1)}{d-1} + 3) + 9 + 4\Delta]$. Then we have

$$\rho^2 = 1 - \frac{1}{ak + b}$$

$$(49)$$

Similar to de Vazelhes et al. (2022), we define:

$$A = \frac{16(1 + \Delta)d\kappa^2(s_2 - 1)}{(s_2 + 2)(d - 1)},$$

$$B = \kappa^2\frac{8(1 + \Delta)d}{(s_2 + 2)}(\frac{(k^* - 1)(s_2 - 1)}{d - 1} + 3),$$

$$C = (9 + 4\Delta)\kappa^2.$$

$$(50)$$

Then we have:

$$\Delta' = 16Ak^*[k^*C(C-1)A + B]$$
$$C = (9 + 4\Delta)\kappa^2 \geq 9 > 1$$
$$\Delta' \geq 0 \tag{51}$$

Case $s_2 > 1$:

$$
\begin{aligned}
q_{min} &= \frac{2Ak^*(2C-1) + \sqrt{16Ak^*[k^*C(C-1)A + B]}}{2} \\
&= \frac{16(1+\Delta)d\kappa^2(s_2-1)k^*}{(s_2+2)(d-1)}[(18+8\Delta)\kappa^2 - 1 \\
&\quad + 2\sqrt{(9+4\Delta)\kappa^2((9+4\Delta)\kappa^2 - 1) + \frac{1}{2} - \frac{1}{2k^*} + \frac{3}{2}\frac{d-1}{k^*(s_2-1)}}]
\end{aligned}
\tag{52}
$$

Case $s_2 = 1$, which leads to $A = 0$, is also similar to de Vazelhes et al. (2022):

$$q \geq \frac{B}{\sqrt{\frac{d}{k^*}} + 1}$$
$$q \geq \frac{8(1+\Delta)\kappa^2 d}{\sqrt{\frac{d}{k^*}} + 1} \tag{53}$$

# D    PROOF OF VARIANCE REDUCTION ALGORITHMS

## D.1    PROOF OF THEOREM 2

**Lemma 4.** For any support $F \in [d]$ with $|F| = s$, assuming that each $f_i$ satisfies $(L_s, s)$-RSS, for the ZO estimator described in out work, with $q$ random directions, and random supports of size $s_2$, we have:

$$
\begin{aligned}
&\mathbb{E}_u\|\hat{\nabla}_F f_{\xi,i}(x) - \nabla_F f_i(x^*)\|^2 \\
&\leq 4(1+\Delta)\varepsilon_F\|\nabla f_i(x) - \nabla f_i(x^*)\|^2 + 2(1+\Delta)\varepsilon_{abs}\mu^2 + 8sd(1+\frac{1}{\Delta})\frac{\Delta^2}{\mu^2} \\
&\quad + ((4(1+\Delta)\varepsilon_F + 2)s + 4(1+\Delta)\varepsilon_{F^c}(d-k))\|\nabla f_i(x^*)\|_\infty^2
\end{aligned}
\tag{54}
$$

$Proof.$

$$
\begin{aligned}
&\mathbb{E}_u\|\hat{\nabla}_F f_{\xi,i}(x) - \nabla_F f_i(x^*)\|^2 \\
&\leq 2\mathbb{E}_u\|\hat{\nabla}_F f_{\xi,i}(x)\|^2 + 2\|\nabla_F f_i(x^*)\|^2 \\
&\overset{(a)}{\leq} 2(1+\Delta)\varepsilon_F\|\nabla_F f_i(x)\|^2 + 2(1+\Delta)\varepsilon_{F^c}\|\nabla_{F^c} f_i(x)\|^2 + 2(1+\Delta)\varepsilon_{abs}\mu^2 + 8sd(1+\frac{1}{\Delta})\frac{\Delta^2}{\mu^2} \\
&\quad + 2\|\nabla_F f_i(x^*)\|^2 \\
&\leq 2(1+\Delta)\varepsilon_F[2\|\nabla_F f_i(x) - \nabla_F f_i(x^*)\|^2 + 2\|\nabla_F f_i(x^*)\|^2] + 2(1+\Delta)\varepsilon_{abs}\mu^2 + 8sd(1+\frac{1}{\Delta})\frac{\Delta^2}{\mu^2} \\
&\quad + 2(1+\Delta)\varepsilon_{F^c}[2\|\nabla_{F^c} f_i(x) - \nabla_{F^c} f_i(x^*)\|^2 + 2\|\nabla_{F^c} f_i(x^*)\|^2] + 2\|\nabla_F f_i(x^*)\|^2 \\
&= 4(1+\Delta)\varepsilon_F\|\nabla_F f_i(x) - \nabla_F f_i(x^*)\|^2 + 4(1+\Delta)\varepsilon_{F^c}\|\nabla_{F^c} f_i(x) - \nabla_{F^c} f_i(x^*)\|^2 \\
&\quad + (4(1+\Delta)\varepsilon_F + 2)\|\nabla_F f_i(x^*)\|^2 + 4(1+\Delta)\varepsilon_{F^c}\|\nabla_{F^c} f_i(x^*)\|^2 + (1+\Delta)2\varepsilon_{abs}\mu^2 \\
&\quad + 8sd(1+\frac{1}{\Delta})\frac{\Delta^2}{\mu^2}
\end{aligned}
\tag{55}
$$

Where (a) follows from Proposition 1(b).

Then we have $\|\nabla_F f_i(x^*)\|_2^2 \leq \|\nabla_s f_i(x^*)\|_2^2 \leq s\|\nabla f_i(x^*)\|_\infty^2$, and $\|\nabla_{F^c} f_i(x^*)\|_2^2 \leq (d-k)\|\nabla f_i(x^*)\|_\infty^2$, since $|F^c| \leq d-k$, to transform $\ell_2$ norm to $\ell_\infty$ norm.

Therefore, we obtain:

$$\mathbb{E}_u \|\hat{\nabla}_F f_{\xi,i}(x) - \nabla_F f_i(x^*)\|^2$$

$$\leq 4(1+\Delta)\varepsilon_F \|\nabla_F f_i(x) - \nabla_F f_i(x^*)\|^2 + 4(1+\Delta)\varepsilon_{F^c}\|\nabla_{F^c} f_i(x) - \nabla_{F^c} f_i(x^*)\|^2$$

$$+ ((4(1+\Delta)\varepsilon_F + 2)s + 4(1+\Delta)\varepsilon_{F^c}(d-k))\|\nabla f_i(x^*)\|_\infty^2 + 2(1+\Delta)\varepsilon_{abs}\mu^2 + 8sd(1+\frac{1}{\Delta})\frac{\Delta^2}{\mu^2}$$

$$\overset{(a)}{\leq} 4(1+\Delta)\varepsilon_F\|\nabla f_i(x) - \nabla f_i(x^*)\|^2 + ((4(1+\Delta)\varepsilon_F + 2)s + 4(1+\Delta)\varepsilon_{F^c}(d-k))\|\nabla f_i(x^*)\|_\infty^2$$

$$+ 2(1+\Delta)\varepsilon_{abs}\mu^2 + 8sd(1+\frac{1}{\Delta})\frac{\Delta^2}{\mu^2}$$

$$(56)$$

Where (a) follows by the fact $\varepsilon_{F^c} \leq \varepsilon_F$ and the definition of the Euclidean norm. Then we can prove Theorem 2, and for the sake of the aesthetics of the article, we have made some modifications to the expression of this formula in the main text.

**Theorem 2.** Suppose $\|x^*\|_0 \leq k^*$, $\mathcal{F}(x)$ satisfies $(m_s, s)$-RSC, the functions $\{f_i(x)\}_{i=1}^n$ satisfies $(L_s, s)$-RSS with $s = 2k + k^*$, and the noisy functions $\{f_{\xi,i}(x)\}_{i=1}^n$ satisfy Assumption 1. Run $\alpha$-VR-SZHT after $m$ iterations, we have:

$$\gamma[\mathcal{F}(x^{(r)}) - \mathcal{F}(x^*)] \leq \rho_\alpha[\mathcal{F}(x^{(r-1)}) - \mathcal{F}(x^*)] + L_\delta + L_{\Delta,\mu},$$

$$with \quad \beta = (1+\eta^2 m_s^2)\lambda, l = \lambda\frac{\beta^m - 1}{\beta - 1},$$

$$\delta'^2 = \mathbb{E}_{i_r}\|\nabla f_{i_r}(x^*)\|_\infty^2,$$

$$\gamma = (2\eta - 48(1+\Delta)\eta^2\varepsilon_F L_s)l,$$

$$\rho_\alpha = 2\frac{\beta^m}{m_s} + 48\alpha^2(1+\Delta)\eta^2 l\varepsilon_F L_s,$$

$$L_\delta = 2\frac{\beta^m}{m_s}\sqrt{s}\delta\mathbb{E}_{u,i_r}\|x^{(r-1)} - x^*\| + 6\eta^2 s((1-\alpha)^2\delta'^2 + \alpha^2\delta^2)l + 3\eta^2(1+\alpha^2)((4\varepsilon_F + 2)s$$

$$+ 4\varepsilon_{F^c}(d-k))\delta'^2 l,$$

$$L_{\Delta,\mu} = 6\eta^2(1+\alpha^2)(\varepsilon_{abs}\mu^2\Delta + 4sd(1+\frac{1}{\Delta})\frac{\Delta^2}{\mu^2})l + 4\frac{sd}{m_s^2}\frac{\Delta^2}{\mu^2}l + (\frac{sdL_s^2}{m_s^2} + 6\eta^2(1+\alpha^2)\varepsilon_{abs})l\mu^2$$

$$+ 3\eta^2(1+\alpha^2)(4\varepsilon_F s + 4\varepsilon_{F^c}(d-k))\delta'^2 l\Delta + 4\frac{sdL_s}{m_s^2}l\Delta,$$

$$(57)$$

$Here \ \varepsilon_\mu = L_s^2 sd, \varepsilon_F = \frac{2d}{q(s_2+2)}(\frac{(s-1)(s_2-1)}{d-1} + 3) + 2, \varepsilon_{F^c} = \frac{2sd(s_2-1)}{q(s_2+2)(d-1)}, \varepsilon_{abs} = \frac{2dL_s^2 ss_2}{q}(\frac{(s-1)(s_2-1)}{d-1} + 1) + L_s^2 sd.$

$Proof.$ First we focus on $\|y^{(t)} - x^*\|$ with variance reduction $y^{(t)} = x^{(t)} - \eta g^{(t)}$, where $g^{(t)} = \hat{\nabla}_F f_{\xi,i_r}(x^{(t)}) - \alpha\hat{\nabla}_F f_{\xi,i_r}(x^{(0)}) + \alpha\hat{\nabla}_F \mathcal{F}_\xi(x^{(0)})$, we can easily note that $\mathbb{E}_{u,i_r}g^{(r)} =$

$\mathbb{E}_u \hat{\nabla}_F \mathcal{F}_\xi(x^{(r)})$, we have:

$$
\begin{aligned}
&\mathbb{E}_{u,i_r} \|y^{(t)} - x^*\|^2 \\
&= \mathbb{E}_{u,i_r} \|x^{(t)} - \eta g^{(t)} - x^*\|^2 \\
&= \mathbb{E}_{u,i_r} \|x^{(t)} - x^*\|^2 - 2\eta \mathbb{E}_{u,i_r} \langle x^{(t)} - x^*, g^{(t)} - \nabla_F \mathcal{F}(x^{(t)}) \rangle - 2\eta \mathbb{E}_{u,i_r} \langle x^{(t)} - x^*, \nabla_F \mathcal{F}(x^{(t)}) \rangle \\
&\quad + \eta^2 \mathbb{E}_{u,i_r} \|g^{(t)}\|^2 \\
&\leq \mathbb{E}_{u,i_r} \|x^{(t)} - x^*\|^2 - 2\eta \langle x^{(t)} - x^*, \mathbb{E}_{u,i_r} g^{(t)} - \nabla_F \mathcal{F}(x^{(t)}) \rangle + \eta^2 E_{u,i_r} \|g^{(t)}\|^2 \\
&\quad - 2\eta (\mathcal{F}(x^{(t)}) - \mathcal{F}(x^*)) \\
&= \mathbb{E}_{u,i_r} \|x^{(t)} - x^*\|^2 - 2\eta \langle x^{(t)} - x^*, \mathbb{E}_u \hat{\nabla}_F \mathcal{F}_\xi(x^{(t)}) - \nabla_F \mathcal{F}(x^{(t)}) \rangle + \eta^2 \mathbb{E}_{u,i_r} \|g^{(t)}\|^2 \\
&\quad - 2\eta (\mathcal{F}(x^{(t)}) - \mathcal{F}(x^*)) \\
&\leq (1 + \eta^2 m_s^2) \mathbb{E}_{u,i_r} \|x^{(t)} - x^*\|^2 + \frac{1}{m_s^2} \|\mathbb{E}_u \hat{\nabla}_F \mathcal{F}_\xi(x^{(t)}) - \nabla_F \mathcal{F}(x^{(t)})\|^2 + \eta^2 E_{u,i_r} \|g^{(t)}\|^2 \\
&\quad - 2\eta (\mathcal{F}(x^{(t)}) - \mathcal{F}(x^*)) \\
&= (1 + \eta^2 m_s^2) \mathbb{E}_{u,i_r} \|x^{(t)} - x^*\|^2 + \frac{1}{m_s^2} (\sqrt{\varepsilon_\mu \mu^2} + 2\frac{\Delta}{\mu}\sqrt{sd})^2 + \eta^2 \mathbb{E}_{u,i_r} \|g^{(t)}\|^2 \\
&\quad - 2\eta (\mathcal{F}(x^{(t)}) - \mathcal{F}(x^*))
\end{aligned}
\tag{58}
$$

For $\mathbb{E}_{u,i_r} \|g^{(t)}\|^2$, we have:

$$
\begin{aligned}
&\mathbb{E}_{u,i_r} \|g^{(t)}\|^2 \\
&= \mathbb{E}_{u,i_r} \|\hat{\nabla}_F f_{\xi,i_r}(x^{(t)}) - \alpha \hat{\nabla}_F f_{\xi,i_r}(x^{(0)}) + \alpha \hat{\nabla}_F \mathcal{F}_\xi(x^{(0)})\|^2 \\
&\leq 3\mathbb{E}_{u,i_r} \|\hat{\nabla}_F f_{\xi,i_r}(x^{(t)}) - \nabla_F f_{i_r}(x^*)\|^2 + 3\mathbb{E}_{i_r} \|(1-\alpha)\nabla_F f_{i_r}(x^*) + \alpha \nabla_F \mathcal{F}(x^*)\|^2 \\
&\quad + 3\alpha^2 \mathbb{E}_{u,i_r} \|\hat{\nabla}_F f_{\xi,i_r}(x^{(0)}) - \nabla_F f_{i_r}(x^*) - \hat{\nabla}_F \mathcal{F}_\xi(x^{(0)}) + \nabla_F \mathcal{F}(x^*)\|^2 \\
&= 3\mathbb{E}_{u,i_r} \|\hat{\nabla}_F f_{\xi,i_r}(x^{(t)}) - \nabla_F f_{i_r}(x^*)\|^2 + 3\mathbb{E}_{i_r} \|(1-\alpha)\nabla_F f_{i_r}(x^*) + \alpha \nabla_F \mathcal{F}(x^*)\|^2 \\
&\quad + 3\alpha^2 \mathbb{E}_u \{ \mathbb{E}_{i_r} \|\hat{\nabla}_F f_{\xi,i_r}(x^{(0)}) - \nabla_F f_{i_r}(x^*) - [\hat{\nabla}_F \mathcal{F}_\xi(x^{(0)}) - \nabla_F \mathcal{F}(x^*)]\|^2 \} \\
&\overset{(a)}{\leq} 3\mathbb{E}_{u,i_r} \|\hat{\nabla}_F f_{\xi,i_r}(x^{(t)}) - \nabla_F f_{i_r}(x^*)\|^2 + 3\mathbb{E}_{i_r} \|(1-\alpha)\nabla_F f_{i_r}(x^*) + \alpha \nabla_F \mathcal{F}(x^*)\|^2 \\
&\quad + 3\alpha^2 \mathbb{E}_u \{ \mathbb{E}_{i_r} \|\hat{\nabla}_F f_{\xi,i_r}(x^{(0)}) - \nabla_F f_{i_r}(x^*)\|^2 \} \\
&= 3\mathbb{E}_{u,i_r} \|\hat{\nabla}_F f_{\xi,i_r}(x^{(t)}) - \nabla_F f_{i_r}(x^*)\|^2 + 3\mathbb{E}_{i_r} \|(1-\alpha)\nabla_F f_{i_r}(x^*) + \alpha \nabla_F \mathcal{F}(x^*)\|^2 \\
&\quad + 3\alpha^2 \mathbb{E}_{u,i_r} \|\hat{\nabla}_F f_{\xi,i_r}(x^{(0)}) - \nabla_F f_{i_r}(x^*)\|^2 \\
&\overset{(b)}{\leq} 12(1+\Delta)\varepsilon_F \mathbb{E}_{i_r} \|\nabla f_{i_r}(x^{(t)}) - \nabla f_{i_r}(x^*)\|^2 + 12\alpha^2(1+\Delta)\varepsilon_F \mathbb{E}_{i_r} \|\nabla f_{i_r}(x^{(0)}) - \nabla f_{i_r}(x^*)\|^2 \\
&\quad + 6(1+\alpha^2)(1+\Delta)\varepsilon_{abs}\mu^2 + 3(1+\alpha^2)((4(1+\Delta)\varepsilon_F + 2)s \\
&\quad + 4(1+\Delta)\varepsilon_{F^c}(d-k))\mathbb{E}_{i_r} \|\nabla f_{i_r}(x^*)\|_\infty^2 + 24(1+\alpha^2)sd(1+\frac{1}{\Delta})\frac{\Delta^2}{\mu^2} \\
&\quad + 3\mathbb{E}_{i_r} \|(1-\alpha)\nabla_F f_{i_r}(x^*) + \alpha \nabla_F \mathcal{F}(x^*)\|^2
\end{aligned}
\tag{59}
$$

Where (a) follows the fact $\mathbb{E}\|X - \mathbb{E}X\|^2 \leq \mathbb{E}\|X\|^2$. And (b) follows Lemma 4.
For $\mathbb{E}_{i_r} \|\nabla f_{i_r}(x) - \nabla f_{i_r}(x^*)\|^2$, according to Li et al. (2016) we can easily get:

$$
\mathbb{E}_{i_r} \|\nabla f_{i_r}(x) - \nabla f_{i_r}(x^*)\|^2 = \frac{1}{n} \sum_{i=1}^{n} \|\nabla f_i(x) - \nabla f_i(x^*)\|^2 \leq 4L_s[\mathcal{F}(x) - \mathcal{F}(x^*)]
\tag{60}
$$

Then we have:

$$
\begin{aligned}
&\mathbb{E}_{u,i_r}\|g^{(t)}\|^2 \\
&\leq 12(1+\Delta)\varepsilon_F\mathbb{E}_{i_r}\|\nabla f_{i_r}(x^{(t)}) - \nabla f_{i_r}(x^*)\|^2 + 12(1+\Delta)\varepsilon_F\mathbb{E}_{i_r}\|\nabla f_{i_r}(x^{(0)}) - \nabla f_{i_r}(x^*)\|^2 \\
&\quad + 12(1+\Delta)\varepsilon_{abs}\mu^2 + 6((4(1+\Delta)\varepsilon_F + 2)s + 4(1+\Delta)\varepsilon_{F^c}(d-k))E_{i_r}\|\nabla f_{i_r}(x^*)\|_\infty^2 \\
&\quad + 48sd(1+\frac{1}{\Delta})\frac{\Delta^2}{\mu^2} + 3\|\nabla_F\mathcal{F}(x^*)\|^2 \\
&\leq 48(1+\Delta)\varepsilon_F L_s[\mathbb{E}\mathcal{F}(x^t) - \mathcal{F}(x^*)] + 48\alpha^2(1+\Delta)\varepsilon_F L_s[\mathcal{F}(x^{(0)}) - \mathcal{F}(x^*)] \\
&\quad + 6(1+\alpha^2)(1+\Delta)\varepsilon_{abs}\mu^2 + 3(1+\alpha^2)((4(1+\Delta)\varepsilon_F + 2)s \\
&\quad + 4(1+\Delta)\varepsilon_{F^c}(d-k))E_{i_r}\|\nabla f_{i_r}(x^*)\|_\infty^2 + 24(1+\alpha^2)sd(1+\frac{1}{\Delta})\frac{\Delta^2}{\mu^2} \\
&\quad + 3E_{i_r}\|(1-\alpha)\nabla_F f_{i_r}(x^*) + \alpha\nabla_F\mathcal{F}(x^*)\|^2
\end{aligned}
\tag{61}
$$

Plug into 58, we get:

$$
\begin{aligned}
&\mathbb{E}_{u,i_r}\|y^{(t)} - x^*\|^2 \\
&\leq (1+\eta^2 m_s^2)\mathbb{E}_{u,i_r}\|x^{(t)} - x^*\|^2 + \frac{1}{m_s^2}(\sqrt{\varepsilon_\mu\mu^2} + 2\frac{\Delta}{\mu}\sqrt{sd})^2 \\
&\quad + (48(1+\Delta)\eta^2\varepsilon_F L_s - 2\eta)[\mathcal{F}(x^t) - \mathcal{F}(x^*)] + 48\alpha^2(1+\Delta)\eta^2\varepsilon_F L_s[\mathcal{F}(x^{(0)}) - \mathcal{F}(x^*)] \\
&\quad + 3\eta^2(1+\alpha^2)((4(1+\Delta)\varepsilon_F + 2)s + 4(1+\Delta)\varepsilon_{F^c}(d-k))E_{i_r}\|\nabla f_{i_r}(x^*)\|_\infty^2 \\
&\quad + \eta^2(24(1+\alpha^2)sd(1+\frac{1}{\Delta})\frac{\Delta^2}{\mu^2} + 3\mathbb{E}_{i_r}\|(1-\alpha)\nabla_F f_{i_r}(x^*) + \alpha\nabla_F\mathcal{F}(x^*)\|^2 \\
&\quad + 6(1+\alpha^2)(1+\Delta)\varepsilon_{abs}\mu^2)
\end{aligned}
\tag{62}
$$

According to 3 and assumption 4, we have:

$$
\begin{aligned}
&\mathbb{E}_{u,i_r}\|x^{(t+1)} - x^*\|^2 \\
&\leq (1+\eta^2 m_s^2)\lambda\mathbb{E}_{u,i_r}\|x^{(t)} - x^*\|^2 + \frac{\lambda}{m_s^2}(\sqrt{\varepsilon_\mu\mu^2} + 2\frac{\Delta}{\mu}\sqrt{sd})^2 \\
&\quad + (48(1+\Delta)\eta^2\varepsilon_F L_s - 2\eta)\lambda[\mathcal{F}(x^{(t)}) - \mathcal{F}(x^*)] + 48(1+\Delta)\alpha^2\eta^2\varepsilon_F L_s\lambda[\mathcal{F}(x^{(0)}) - \mathcal{F}(x^*)] \\
&\quad + 3\eta^2\lambda(1+\alpha^2)((4(1+\Delta)\varepsilon_F + 2)s + 4(1+\Delta)\varepsilon_{F^c}(d-k))\mathbb{E}_{i_r}\|\nabla f_{i_r}(x^*)\|_\infty^2 \\
&\quad + \eta^2\lambda(24(1+\alpha^2)sd(1+\frac{1}{\Delta})\frac{\Delta^2}{\mu^2} + 3\mathbb{E}_{i_r}\|(1-\alpha)\nabla_F f_{i_r}(x^*) + \alpha\nabla_F\mathcal{F}(x^*)\|^2) \\
&\quad + 6\eta^2\lambda(1+\alpha^2)(1+\Delta)\varepsilon_{abs}\mu^2
\end{aligned}
\tag{63}
$$

We can define:

$$\beta := (1 + \eta^2 m_s^2)\lambda$$

$$M := \frac{1}{m_s^2}(\sqrt{\varepsilon_\mu \mu^2} + 2\frac{\Delta}{\mu}\sqrt{sd})^2 + 6\eta^2(1+\alpha^2)(1+\Delta)\varepsilon_{abs}\mu^2$$

$$+ 3\eta^2(1+\alpha^2)((4(1+\Delta)\varepsilon_F + 2)s + 4(1+\Delta)\varepsilon_{F^c}(d-k))\mathbb{E}_{i_r}\|\nabla f_{i_r}(x^*)\|_\infty^2$$

$$+ \eta^2(24(1+\alpha^2)sd(1+\frac{1}{\Delta})\frac{\Delta^2}{\mu^2} + 3\mathbb{E}_{i_r}\|(1-\alpha)\nabla_F f_{i_r}(x^*) + \alpha\nabla_F\mathcal{F}(x^*)\|^2)$$

$$= (\frac{sdL_s^2}{m_s^2} + 6\eta^2(1+\alpha^2)\varepsilon_{abs})\mu^2$$

$$+ (4\frac{sdL_s}{m_s^2} + 3\eta^2(1+\alpha^2)(4\varepsilon_F s + 4\varepsilon_{F^c}(d-k))\mathbb{E}_{i_r}\|\nabla f_{i_r}(x^*)\|_\infty^2)\Delta$$

$$+ 6\eta^2(1+\alpha^2)\varepsilon_{abs}\mu^2\Delta + 24sd\eta^2(1+\alpha^2)(1+\frac{1}{\Delta})\frac{\Delta^2}{\mu^2}$$

$$+ 3\eta^2\mathbb{E}_{i_r}\|(1-\alpha)\nabla_F f_{i_r}(x^*) + \alpha\nabla_F\mathcal{F}(x^*)\|^2$$

$$+ 3\eta^2(1+\alpha^2)((4\varepsilon_F + 2)s + 4\varepsilon_{F^c}(d-k))\mathbb{E}_{i_r}\|\nabla f_{i_r}(x^*)\|_\infty^2 + 4\frac{sd}{m_s^2}\frac{\Delta^2}{\mu^2} \tag{64}$$

Then we have:

$$\mathbb{E}_{u,i_r}\|x^{(t+1)} - x^*\|^2 + (2\eta - 48(1+\Delta)\eta^2\varepsilon_F L_s)\lambda[\mathcal{F}(x^{(t)}) - \mathcal{F}(x^*)]$$

$$\leq \beta\mathbb{E}_{u,i_r}\|x^{(t)} - x^*\|^2 + 48\alpha^2(1+\Delta)\eta^2\varepsilon_F L_s\lambda[\mathcal{F}(x^{(0)}) - \mathcal{F}(x^*)] + \lambda M \tag{65}$$

By summing over $t = 0, \ldots, m-1$, we have:

$$\mathbb{E}_{u,i_r}\|x^{(m)} - x^*\|^2 + \frac{\beta^m - 1}{\beta - 1}(2\eta - 48(1+\Delta)\eta^2\varepsilon_F L_s)\lambda[\mathbb{E}\mathcal{F}(x^{(r+1)}) - \mathcal{F}(x^*)]$$

$$\leq \beta^m\mathbb{E}_{u,i_r}\|x^{(0)} - x^*\|^2 + \lambda M\frac{\beta^m - 1}{\beta - 1} + 48\alpha^2(1+\Delta)\eta^2\varepsilon_F L_s\lambda\frac{\beta^m - 1}{\beta - 1}[\mathbb{E}\mathcal{F}(x^{(r)}) - \mathcal{F}(x^*)] \tag{66}$$

Through RSC condition ,and the definition of $F$ ,we can get:

$$\frac{\beta^m - 1}{\beta - 1}(2\eta - 48(1+\Delta)\eta^2\varepsilon_F L_s)\lambda[\mathbb{E}\mathcal{F}(x^{(r+1)}) - \mathcal{F}(x^*)]$$

$$\leq (2\frac{\beta^m}{m_s} + 48\alpha^2(1+\Delta)\eta^2\varepsilon_F L_s\lambda\frac{\beta^m - 1}{\beta - 1})[\mathbb{E}\mathcal{F}(x^{(r)}) - \mathcal{F}(x^*)]$$

$$+ 2\frac{\beta^m}{m_s}\mathbb{E}_{u,i_r}\langle\nabla\mathcal{F}(x^*), x^{(r)} - x^*\rangle + \lambda M\frac{\beta^m - 1}{\beta - 1}$$

$$\leq (2\frac{\beta^m}{m_s} + 48\alpha^2(1+\Delta)\eta^2\varepsilon_F L_s\lambda\frac{\beta^m - 1}{\beta - 1})[\mathbb{E}\mathcal{F}(x^{(r)}) - \mathcal{F}(x^*)]$$

$$+ 2\frac{\beta^m}{m_s}\sqrt{s}\delta\mathbb{E}_{u,i_r}\|x^{(r)} - x^*\| + \lambda M\frac{\beta^m - 1}{\beta - 1} \tag{67}$$

**Corollary 2.** To converge, we need:

$$\Delta < \frac{\eta(\beta^m - 1)\lambda m_s - (\beta - 1)\beta^m}{24\eta^2(\beta^m - 1)\lambda m_s\varepsilon_F L_s(\alpha^2 + 1)} - 1 \tag{68}$$

Large $\alpha$ decrease the tolerable upper bound of noise.

*Proof.* If algorithm converges, we need:

$$2\frac{\beta^m}{m_s} + 48\alpha^2(1+\Delta)\eta^2\varepsilon_F L_s\lambda\frac{\beta^m - 1}{\beta - 1} \leq \frac{\beta^m - 1}{\beta - 1}(2\eta - 48(1+\Delta)\eta^2\varepsilon_F L_s)\lambda \tag{69}$$

We can get:

$$24(1+\alpha^2)(1+\Delta)\eta^2\varepsilon_F L_s\lambda\frac{\beta^m-1}{\beta-1} \leq \frac{\eta(\beta^m-1)\lambda m_s - \beta^m(\beta-1)}{\beta-1} \tag{70}$$

Then we have:

$$\Delta < \frac{\eta(\beta^m-1)\lambda m_s - (\beta-1)\beta^m}{24\eta^2(\beta^m-1)\lambda m_s\varepsilon_F L_s(\alpha^2+1)} - 1 \tag{71}$$

### D.2 PROOF OF THEOREM 3

**Theorem 3.** Suppose $\|x^*\|_0 \leq k^*$, $\mathcal{F}(x)$ satisfies $(m_s, s)$-RSC, the functions $\{f_i(x)\}_{i=1}^n$ satisfies $(L_s, s)$-RSS with $s = 2k + k^*$, and the noisy functions $\{f_{\xi,i}(x)\}_{i=1}^n$ satisfy Assumption 1. Run $\alpha$-$p$M-SZHT at $r$-th iteration, we have:

$$\mathcal{F}(x^{(r)}) - \mathcal{F}(x^*) \leq \rho'_\alpha(\mathcal{F}(x^{(r-1)}) - \mathcal{F}(x^*)) + L'_\delta + L'_{\Delta,\mu},$$

$$with \quad \beta = (1+\eta^2 m_s^2)\lambda,$$

$$\rho'_\alpha = \frac{2\beta}{m_s} + 48\eta^2\lambda(1+\Delta)\varepsilon_F L_s - 2\eta\lambda + 2\alpha^2(1+\Delta)(1-\frac{p}{n}),$$

$$\delta'^2 = \mathbb{E}_{i_r}\|\nabla f_{i_r}(x^*)\|_\infty^2,$$

$$L'_\delta = 6\eta^2\lambda s((1-\alpha)^2\delta'^2 + \alpha^2\delta^2),$$

$$L'_{\Delta,\mu} = \eta^2\lambda(6(1+\Delta)\varepsilon_{abs}\mu^2$$

$$+ 24sd(1+\frac{1}{\Delta})\frac{\Delta^2}{\mu^2}) + 3\eta^2\lambda\alpha^2 A_{r-1} + \frac{\lambda}{m_s^2}(\sqrt{\varepsilon_\mu\mu^2} + 2\frac{\Delta}{\mu}\sqrt{sd})^2 + 3\eta^2\lambda\tau\delta'^2,$$

$$\tau = (4(1+\Delta)\varepsilon_F + 2)s + 4(1+\Delta)\varepsilon_{F^c}(d-k). \tag{72}$$

$Here\ \varepsilon_\mu = L_s^2 sd, \varepsilon_F = \frac{2d}{q(s_2+2)}(\frac{(s-1)(s_2-1)}{d-1} + 3) + 2, \varepsilon_{F^c} = \frac{2sd(s_2-1)}{q(s_2+2)(d-1)}, \varepsilon_{abs} = \frac{2dL_s^2 ss_2}{q}(\frac{(s-1)(s_2-1)}{d-1} + 1) + L_s^2 sd.$

$Proof.$ First we focus on $\|y^{(r)} - x^*\|$ with variance reduction $y^{(r)} = x^{(r)} - \eta g^{(r)}$, $g^{(r)} = \hat{\nabla}_F f_{\xi,i_r}(x^{(r)}) - \alpha(\hat{a}_{\xi,i_r}^{(r)} - \mathbb{E}_{i_r}\hat{a}_{\xi,i_r}^{(r)})$, we can easily note that $\mathbb{E}_{u,i_r}g^{(r)} = \mathbb{E}_u\hat{\nabla}_F\mathcal{F}_\xi(x^{(r)})$. We have:

$$\mathbb{E}_{u,i_r}\|y^{(r)} - x^*\|^2$$

$$= \mathbb{E}_{u,i_r}\|x^{(r)} - \eta g^{(r)} - x^*\|^2$$

$$= \mathbb{E}_{u,i_r}\|x^{(r)} - x^*\|^2 - 2\eta\mathbb{E}_{u,i_r}\langle x^{(r)} - x^*, g^{(r)} - \nabla_F\mathcal{F}(x^{(r)})\rangle$$

$$\quad - 2\eta\mathbb{E}_{u,i_r}\langle x^{(r)} - x^*, \nabla_F\mathcal{F}(x^{(r)})\rangle + \eta^2\mathbb{E}_{u,i_r}\|g^{(r)}\|^2$$

$$\leq \mathbb{E}_{u,i_r}\|x^{(r)} - x^*\|^2 - 2\eta\langle x^{(r)} - x^*, \mathbb{E}_{u,i_r}g^{(r)} - \nabla_F\mathcal{F}(x^{(r)})\rangle + \eta^2\mathbb{E}_{u,i_r}\|g^{(r)}\|^2$$

$$\quad - 2\eta(\mathcal{F}(x^{(r)}) - \mathcal{F}(x^*))$$

$$= \mathbb{E}_{u,i_r}\|x^{(r)} - x^*\|^2 - 2\eta\langle x^{(r)} - x^*, \mathbb{E}_u\hat{\nabla}_F\mathcal{F}_\xi(x^{(r)}) - \nabla_F\mathcal{F}(x^{(r)})\rangle + \eta^2\mathbb{E}_{u,i_r}\|g^{(r)}\|^2$$

$$\quad - 2\eta(\mathcal{F}(x^{(r)}) - \mathcal{F}(x^*))$$

$$\leq (1+\eta^2 m_s^2)\mathbb{E}_{u,i_r}\|x^{(r)} - x^*\|^2 + \frac{1}{m_s^2}\|\mathbb{E}_u\hat{\nabla}_F\mathcal{F}_\xi(x^{(r)}) - \nabla_F\mathcal{F}(x^{(r)})\|^2 + \eta^2\mathbb{E}_{u,i_r}\|g^{(r)}\|^2$$

$$\quad - 2\eta(\mathcal{F}(x^{(r)}) - \mathcal{F}(x^*))$$

$$= (1+\eta^2 m_s^2)\mathbb{E}_{u,i_r}\|x^{(r)} - x^*\|^2 + \frac{1}{m_s^2}(\sqrt{\varepsilon_\mu\mu^2} + 2\frac{\Delta}{\mu}\sqrt{sd})^2 + \eta^2\mathbb{E}_{u,i_r}\|g^{(r)}\|^2$$

$$\quad - 2\eta(\mathcal{F}(x^{(r)}) - \mathcal{F}(x^*))$$

$$\tag{73}$$

For $\mathbb{E}_{u,i_r}\|g^{(r)}\|^2$, we have:

$$
\begin{aligned}
&\mathbb{E}_{u,i_r}\|g^{(r)}\|^2 \\
&= \mathbb{E}_{u,i_r}\|\hat{\nabla}_F f_{\xi,i_r}(x^{(r)}) - \alpha(\hat{a}_{\xi,i_r}^{(r)} - \mathbb{E}_{i_r}\hat{a}_{\xi,i_r}^{(r)})\|^2 \\
&= \mathbb{E}_{u,i_r}\|\hat{\nabla}_F f_{\xi,i_r}(x^{(r)}) - \hat{\nabla}_F f_{i_r}(x^*) - \alpha(\hat{a}_{\xi,i_r}^{(r)} - \mathbb{E}_{i_r}\hat{a}_{\xi,i_r}^{(r)} - \hat{\nabla}_F f_{i_r}(x^*) + \hat{\nabla}_F \mathcal{F}(x^*)) \\
&\quad + (1-\alpha)\hat{\nabla}_F f_{i_r}(x^*) + \alpha\hat{\nabla}_F \mathcal{F}(x^*)\|^2 \\
&\leq 3\mathbb{E}_{u,i_r}\|\hat{\nabla}_F f_{\xi,i_r}(x^{(r)}) - \nabla_F f_{i_r}(x^*)\|^2 + 3\mathbb{E}_{i_r}\|(1-\alpha)\nabla_F f_{i_r}(x^*) + \alpha\nabla_F \mathcal{F}(x^*)\|^2 \\
&\quad + 3\alpha^2\mathbb{E}_{u,i_r}\|\hat{a}_{\xi,i_r}^{(r)} - \mathbb{E}_{i_r}\hat{a}_{\xi,i_r}^{(r)} - \hat{\nabla}_F f_{i_r}(x^*) + \hat{\nabla}_F \mathcal{F}(x^*)\|^2 \\
&= 3\mathbb{E}_{u,i_r}\|\hat{\nabla}_F f_{\xi,i_r}(x^{(r)}) - \nabla_F f_{i_r}(x^*)\|^2 + 3\mathbb{E}_{i_r}\|(1-\alpha)\nabla_F f_{i_r}(x^*) + \alpha\nabla_F \mathcal{F}(x^*)\|^2 \\
&\quad + 3\alpha^2\mathbb{E}_u\{\mathbb{E}_{i_r}\|\hat{a}_{\xi,i_r}^{(r)} - \hat{\nabla}_F f_{i_r}(x^*) - (\mathbb{E}_{i_r}\hat{a}_{\xi,i_r}^{(r)} - \hat{\nabla}_F \mathcal{F}(x^*))\|^2\} \\
&\overset{(a)}{\leq} 3\mathbb{E}_{u,i_r}\|\hat{\nabla}_F f_{\xi,i_r}(x^{(r)}) - \nabla_F f_{i_r}(x^*)\|^2 + 3\mathbb{E}_{i_r}\|(1-\alpha)\nabla_F f_{i_r}(x^*) + \alpha\nabla_F \mathcal{F}(x^*)\|^2 \\
&\quad + 3\alpha^2\mathbb{E}_u\{\mathbb{E}_{i_r}\|\hat{a}_{\xi,i_r}^{(r)} - \hat{\nabla}_F f_{i_r}(x^*)\|^2\} \\
&= 3\mathbb{E}_{u,i_r}\|\hat{\nabla}_F f_{\xi,i_r}(x^{(r)}) - \nabla_F f_{i_r}(x^*)\|^2 + 3\mathbb{E}_{i_r}\|(1-\alpha)\nabla_F f_{i_r}(x^*) + \alpha\nabla_F \mathcal{F}(x^*)\|^2 \\
&\quad + 3\alpha^2\mathbb{E}_{u,i_r}\|\hat{a}_{\xi,i_r}^{(r)} - \hat{\nabla}_F f_{i_r}(x^*)\|^2
\end{aligned}
\tag{74}
$$

Where (a) follows the fact $\mathbb{E}\|X - \mathbb{E}X\|^2 \leq \mathbb{E}\|X\|^2$.

For $\mathbb{E}_{u,i_r}\|\hat{a}_{\xi,i_r}^{(r)} - \hat{\nabla}_F f_{i_r}(x^*)\|^2$, from Gu et al. (2020) Lemma 14, we know:

$$
\begin{aligned}
&\mathbb{E}_{u,i_r}\|\hat{a}_{\xi,i_r}^{(r)} - \hat{\nabla}_F f_{i_r}(x^*)\|^2 \\
&\leq \frac{1}{n}\sum_{k=1}^{r-1}\left(1-\frac{p}{n}\right)^{r-k-1}\mathbb{E}_{u,i_r}\|\nabla_F f_{\xi,i_r}(x^{(k)}) - \nabla_F f_{i_r}(x^*)\|^2 \\
&\quad + \left(1-\frac{p}{n}\right)^r\mathbb{E}_{u,i_r}\|\nabla_F f_{\xi,i_r}(x^{(0)}) - \nabla_F f_{i_r}(x^*)\|^2
\end{aligned}
\tag{75}
$$

Where $k$ denote the time of the iterate last used to write the $[\hat{a}_{\xi,i}^{(r)}]$.

Using Lemma 4, we get:

$$
\begin{aligned}
&\mathbb{E}_{u,i_r}\|\hat{a}_{\xi,i_r}^{(r)} - \hat{\nabla}_F f_{i_r}(x^*)\|^2 \\
&\leq 4(1+\Delta)\varepsilon_F\frac{1}{n}\sum_{k=1}^{r-1}\left(1-\frac{p}{n}\right)^{r-k-1}\mathbb{E}_{i_r}\|\nabla f_{i_r}(x^{(k)}) - \nabla f_{i_r}(x^*)\|^2 \\
&\quad + 4(1+\Delta)\varepsilon_F\left(1-\frac{p}{n}\right)^r\mathbb{E}_{i_r}\|\nabla f_{i_r}(x^{(0)}) - \nabla f_{i_r}(x^*)\|^2 \\
&\quad + \frac{1}{n}\sum_{k=1}^{r-1}\left(1-\frac{p}{n}\right)^{r-k-1}(2(1+\Delta)\varepsilon_{abs}\mu^2 + 8sd(1+\frac{1}{\Delta})\frac{\Delta^2}{\mu^2}) \\
&\quad + \frac{1}{n}\sum_{k=1}^{r-1}\left(1-\frac{p}{n}\right)^{r-k-1}((4(1+\Delta)\varepsilon_F + 2)s + 4(1+\Delta)\varepsilon_{F^c}(d-k))\mathbb{E}_{i_r}\|\nabla f_i(x^*)\|_\infty^2 \\
&\quad + \left(1-\frac{p}{n}\right)^r(2(1+\Delta)\varepsilon_{abs}\mu^2 + 8sd(1+\frac{1}{\Delta})\frac{\Delta^2}{\mu^2}) \\
&\quad + \left(1-\frac{p}{n}\right)^r((4(1+\Delta)\varepsilon_F + 2)s + 4(1+\Delta)\varepsilon_{F^c}(d-k))\mathbb{E}_{i_r}\|\nabla f_i(x^*)\|_\infty^2
\end{aligned}
\tag{76}
$$

By the fact that $\mathbb{E}_{i_r}\|\nabla f_{i_r}(x) - \nabla f_{i_r}(x^*)\|^2 = \frac{1}{n}\sum_{i=1}^{n}\|\nabla f_i(x) - \nabla f_i(x^*)\|^2 \leq 4L_s[\mathcal{F}(x) - \mathcal{F}(x^*)]$, we have:

$$
\mathbb{E}_{u,i_r}\|\hat{a}_{\xi,i_r}^{(r)} - \hat{\nabla}_F f_{i_r}(x^*)\|^2
$$

$$
\leq 16(1+\Delta)\varepsilon_F L_s \frac{1}{n}\sum_{k=1}^{r-1}\left(1-\frac{p}{n}\right)^{r-k-1}(\mathcal{F}(x^{(k)}) - \mathcal{F}(x^*))
$$

$$
+ 16(1+\Delta)\varepsilon_F L_s \left(1-\frac{p}{n}\right)^r (\mathcal{F}(x^{(0)}) - \mathcal{F}(x^*))
$$

$$
+ \frac{1}{n}\sum_{k=1}^{r-1}\left(1-\frac{p}{n}\right)^{r-k-1}(2(1+\Delta)\varepsilon_{abs}\mu^2 + 8sd(1+\frac{1}{\Delta})\frac{\Delta^2}{\mu^2})
$$

$$
+ \frac{1}{n}\sum_{k=1}^{r-1}\left(1-\frac{p}{n}\right)^{r-k-1}((4(1+\Delta)\varepsilon_F + 2)s + 4(1+\Delta)\varepsilon_{F^c}(d-k))\mathbb{E}_{i_r}\|\nabla f_i(x^*)\|_\infty^2
$$

$$
+ \left(1-\frac{p}{n}\right)^r(2(1+\Delta)\varepsilon_{abs}\mu^2 + 8sd(1+\frac{1}{\Delta})\frac{\Delta^2}{\mu^2})
$$

$$
+ \left(1-\frac{p}{n}\right)^r((4(1+\Delta)\varepsilon_F + 2)s + 4(1+\Delta)\varepsilon_{F^c}(d-k))\mathbb{E}_{i_r}\|\nabla f_i(x^*)\|_\infty^2
$$

$$\tag{77}$$

And let

$$
A_r = \frac{1}{n}\sum_{k=1}^{r-1}\left(1-\frac{p}{n}\right)^{r-k-1}(2(1+\Delta)\varepsilon_{abs}\mu^2 + 8sd(1+\frac{1}{\Delta})\frac{\Delta^2}{\mu^2})
$$

$$
+ \frac{1}{n}\sum_{k=1}^{r-1}\left(1-\frac{p}{n}\right)^{r-k-1}((4(1+\Delta)\varepsilon_F + 2)s + 4(1+\Delta)\varepsilon_{F^c}(d-k))\mathbb{E}_{i_r}\|\nabla f_i(x^*)\|_\infty^2
$$

$$
+ \left(1-\frac{p}{n}\right)^r(2(1+\Delta)\varepsilon_{abs}\mu^2 + 8sd(1+\frac{1}{\Delta})\frac{\Delta^2}{\mu^2})
$$

$$
+ \left(1-\frac{p}{n}\right)^r((4(1+\Delta)\varepsilon_F + 2)s + 4(1+\Delta)\varepsilon_{F^c}(d-k))\mathbb{E}_{i_r}\|\nabla f_i(x^*)\|_\infty^2
$$

$$\tag{78}$$

And for $\mathbb{E}_{u,i_r}\|\hat{\nabla}_F f_{\xi,i_r}(x^{(r)}) - \nabla_F f_{i_r}(x^*)\|^2$, using Lemma 4, we get:

$$
\mathbb{E}_{u,i_r}\|\hat{\nabla}_F f_{\xi,i_r}(x^{(r)}) - \nabla_F f_{i_r}(x^*)\|^2
$$

$$
\leq 4(1+\Delta)\varepsilon_F \mathbb{E}_{i_r}\|\nabla f_{i_r}(x^{(r)}) - \nabla f_{i_r}(x^*)\|^2
$$

$$
+ ((4(1+\Delta)\varepsilon_F + 2)s + 4(1+\Delta)\varepsilon_{F^c}(d-k))\mathbb{E}_{i_r}\|\nabla f_{i_r}(x^*)\|_\infty^2
$$

$$
+ 2(1+\Delta)\varepsilon_{abs}\mu^2 + 8sd(1+\frac{1}{\Delta})\frac{\Delta^2}{\mu^2}
$$

$$
\leq 16(1+\Delta)\varepsilon_F L_s(\mathcal{F}(x^{(r)}) - \mathcal{F}(x^*))
$$

$$
+ ((4(1+\Delta)\varepsilon_F + 2)s + 4(1+\Delta)\varepsilon_{F^c}(d-k))\mathbb{E}_{i_r}\|\nabla f_{i_r}(x^*)\|_\infty^2
$$

$$
+ 2(1+\Delta)\varepsilon_{abs}\mu^2 + 8sd(1+\frac{1}{\Delta})\frac{\Delta^2}{\mu^2}
$$

$$\tag{79}$$

Plugging into 74, we have:

$$\mathbb{E}_{u,i_r}\|g^{(r)}\|^2$$

$$\leq 48(1+\Delta)\varepsilon_F L_s(\mathcal{F}(x^{(r)}) - \mathcal{F}(x^*))$$

$$+ 3((4(1+\Delta)\varepsilon_F + 2)s + 4(1+\Delta)\varepsilon_{F^c}(d-k))\mathbb{E}_{i_r}\|\nabla f_{i_r}(x^*)\|_\infty^2$$

$$+ 6(1+\Delta)\varepsilon_{abs}\mu^2 + 24sd(1+\frac{1}{\Delta})\frac{\Delta^2}{\mu^2} + 3\alpha^2 A_r + 3\mathbb{E}_{i_r}\|(1-\alpha)\nabla_F f_{i_r}(x^*) + \alpha\nabla_F\mathcal{F}(x^*)\|^2$$

$$+ 48\alpha^2(1+\Delta)\varepsilon_F L_s\frac{1}{n}\sum_{k=1}^{r-1}\left(1-\frac{p}{n}\right)^{r-k-1}(\mathcal{F}(x^{(k)}) - \mathcal{F}(x^*))$$

$$+ 48\alpha^2(1+\Delta)\varepsilon_F L_s\left(1-\frac{p}{n}\right)^r(\mathcal{F}(x^{(0)}) - \mathcal{F}(x^*))$$

$$\leq 48(1+\Delta)\varepsilon_F L_s(\mathcal{F}(x^{(r)}) - \mathcal{F}(x^*))$$

$$+ 3((4(1+\Delta)\varepsilon_F + 2)s + 4(1+\Delta)\varepsilon_{F^c}(d-k))\mathbb{E}_{i_r}\|\nabla f_{i_r}(x^*)\|_\infty^2$$

$$+ 6(1+\Delta)\varepsilon_{abs}\mu^2 + 24sd(1+\frac{1}{\Delta})\frac{\Delta^2}{\mu^2} + 3\alpha^2 A_r + 6s((1-\alpha)^2\mathbb{E}_{i_r}\|\nabla f_{i_r}(x^*)\|^2 + \alpha^2\delta^2)$$

$$+ 48\alpha^2(1+\Delta)\varepsilon_F L_s\frac{1}{n}\sum_{k=1}^{r-1}\left(1-\frac{p}{n}\right)^{r-k-1}(\mathcal{F}(x^{(k)}) - \mathcal{F}(x^*))$$

$$+ 48\alpha^2(1+\Delta)\varepsilon_F L_s\left(1-\frac{p}{n}\right)^r(\mathcal{F}(x^{(0)}) - \mathcal{F}(x^*))$$

$$(80)$$

Take 80 into 73, Let $\beta = (1+\eta^2 m_s^2)\lambda$ and

$$M_r = 3\eta^2((4(1+\Delta)\varepsilon_F + 2)s + 4(1+\Delta)\varepsilon_{F^c}(d-k))\mathbb{E}_{i_r}\|\nabla f_{i_r}(x^*)\|_\infty^2$$

$$+ \frac{1}{m_s^2}(\sqrt{\varepsilon_\mu\mu^2} + 2\frac{\Delta}{\mu}\sqrt{sd})^2 + \eta^2(6(1+\Delta)\varepsilon_{abs}\mu^2 + 24sd(1+\frac{1}{\Delta})\frac{\Delta^2}{\mu^2} + 3\alpha^2 A_r$$

$$+ 6s((1-\alpha)^2\mathbb{E}_{i_r}\|\nabla f_{i_r}(x^*)\|^2 + \alpha^2\delta^2))$$

$$(81)$$

Then use RSC and RSS condition, we get:

$$\mathbb{E}\mathcal{F}(x^{(r+1)}) - \mathcal{F}(x^*)$$

$$\leq (\frac{2\beta}{m_s} + 48\eta^2\lambda(1+\Delta)\varepsilon_F L_s - 2\eta\lambda)(\mathbb{E}\mathcal{F}(x^{(r)}) - \mathcal{F}(x^*)) + \lambda M_r$$

$$+ 48\eta^2\alpha^2\lambda(1+\Delta)\varepsilon_F L_s\frac{1}{n}\sum_{k=1}^{r-1}\left(1-\frac{p}{n}\right)^{r-k-1}(\mathcal{F}(x^{(k)}) - \mathcal{F}(x^*))$$

$$+ 48\eta^2\alpha^2\lambda(1+\Delta)\varepsilon_F L_s\left(1-\frac{p}{n}\right)^r(\mathcal{F}(x^{(0)}) - \mathcal{F}(x^*))$$

$$(82)$$

Using the inequalities (57) and (58) of Gu et al. (2020), we get:

$$\mathbb{E}\mathcal{F}(x^{(r+1)}) - \mathcal{F}(x^*)$$

$$\leq (\frac{2\beta}{m_s} + 48\eta^2\lambda(1+\Delta)\varepsilon_F L_s - 2\eta\lambda + 2\alpha^2(1+\Delta)(1-\frac{p}{n}))(\mathbb{E}\mathcal{F}(x^{(r)}) - \mathcal{F}(x^*))$$

$$+ 3\eta^2\lambda((4(1+\Delta)\varepsilon_F + 2)s + 4(1+\Delta)\varepsilon_{F^c}(d-k))\mathbb{E}_{i_r}\|\nabla f_{i_r}(x^*)\|_\infty^2 + \frac{\lambda}{m_s^2}(\sqrt{\varepsilon_\mu\mu^2} + 2\frac{\Delta}{\mu}\sqrt{sd})^2$$

$$+ \eta^2\lambda(6(1+\Delta)\varepsilon_{abs}\mu^2 + 24sd(1+\frac{1}{\Delta})\frac{\Delta^2}{\mu^2} + 3\alpha^2 A_r + 6s((1-\alpha)^2\mathbb{E}_{i_r}\|\nabla f_{i_r}(x^*)\|^2 + \alpha^2\delta^2))$$

$$(83)$$

**Corollary 3.** To converge, we need:

$$\Delta < \frac{(\frac{1}{2} + \eta\lambda)m_s - \beta}{m_s(24\eta^2\lambda\varepsilon_F L_s + \alpha^2(1 - \frac{p}{n}))} - 1 \tag{84}$$

Large $\alpha$ decrease the tolerable upper bound of noise.

$Proof$. If algorithm converges, we need:

$$\frac{2\beta}{m_s} + 48\eta^2\lambda(1+\Delta)\varepsilon_F L_s - 2\eta\lambda + 2\alpha^2(1+\Delta)(1 - \frac{p}{n}) \leq 1 \tag{85}$$

We can get:

$$(24\eta^2\lambda\varepsilon_F L_s + \alpha^2(1 - \frac{p}{n}))(1+\Delta) \leq \frac{(\frac{1}{2} + \eta\lambda)m_s - \beta}{m_s} \tag{86}$$

Then we have:

$$\Delta < \frac{(\frac{1}{2} + \eta\lambda)m_s - \beta}{m_s(24\eta^2\lambda\varepsilon_F L_s + \alpha^2(1 - \frac{p}{n}))} - 1 \tag{87}$$

### D.3 PROOF OF 4.3

$$\|\hat{g}_\xi - \nabla\mathcal{F}(x)\|^2 = \|\hat{g} - \nabla\mathcal{F}(x) + \frac{(1+2\alpha)d}{q}\sum_{i=1}^{q}\frac{\xi_{i1} - \xi_{i2}}{\mu}u_F\|^2$$

$$\leq (1+\Delta)\|\hat{g} - \nabla\mathcal{F}(x)\|^2 + 4sd(1 + \frac{1}{\Delta})(1+2\alpha)^2\frac{\Delta^2}{\mu^2}$$

$$\leq (1+\Delta)\|\hat{\nabla}f_i(x) - \nabla\mathcal{F}(x)\|^2 + 4sd(1 + \frac{1}{\Delta})(1+2\alpha)^2\frac{\Delta^2}{\mu^2} \tag{88}$$

## E  MORE EXPERIMENTS RESULTS

### E.1  SENSITIVITY ANALYSIS

To obtain more universally applicable experimental results, we varied different parameters in the sensitivity analysis experiment, and the outcomes are presented as follows. At first, we keep the original parameter settings of the experiment and change the learning rate $\eta$.

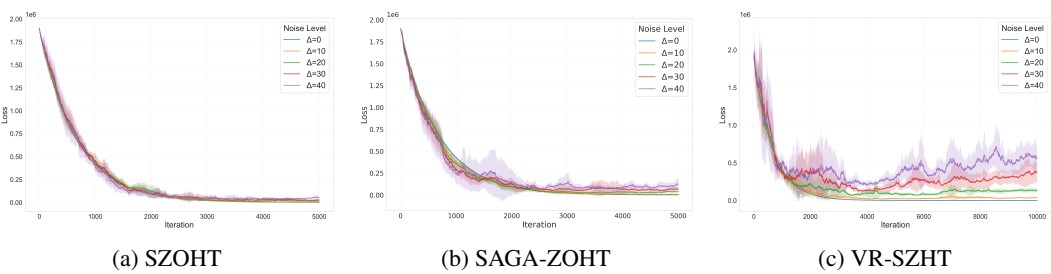

(a) SZOHT  (b) SAGA-ZOHT  (c) VR-SZHT

Figure 4: Set $\eta = 5 \times 10^{-7}, n = 100, d = 100, k = 30, q = 20, \mu = 10^{-4}, m = 5$.

Then we vary the sparsity by change $k$.

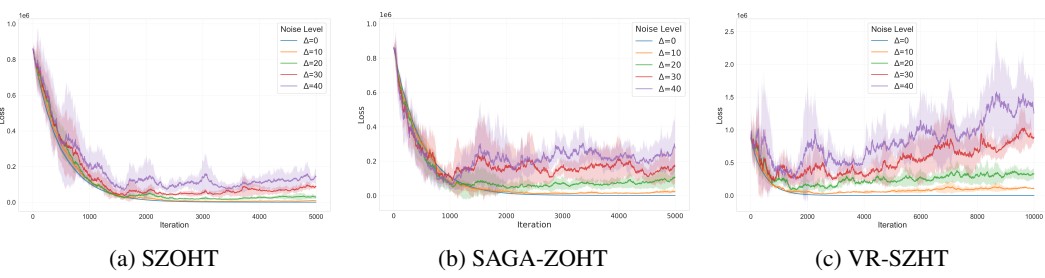

(a) SZOHT          (b) SAGA-ZOHT          (c) VR-SZHT

Figure 5: Set $\eta = 10^{-6}, n = 100, d = 100, k = 20, q = 20, \mu = 10^{-4}, m = 5$.

We reduce the number of random unit vectors $q$ to simulate the case of limited queries.

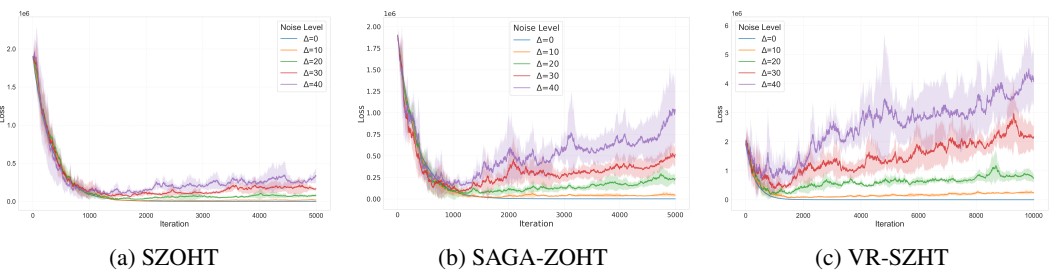

(a) SZOHT          (b) SAGA-ZOHT          (c) VR-SZHT

Figure 6: Set $\eta = 10^{-6}, n = 100, d = 100, k = 30, q = 10, \mu = 10^{-4}, m = 5$.

We also change the update frequency $m$ of VR-SZHT to compare with figure 1(c).

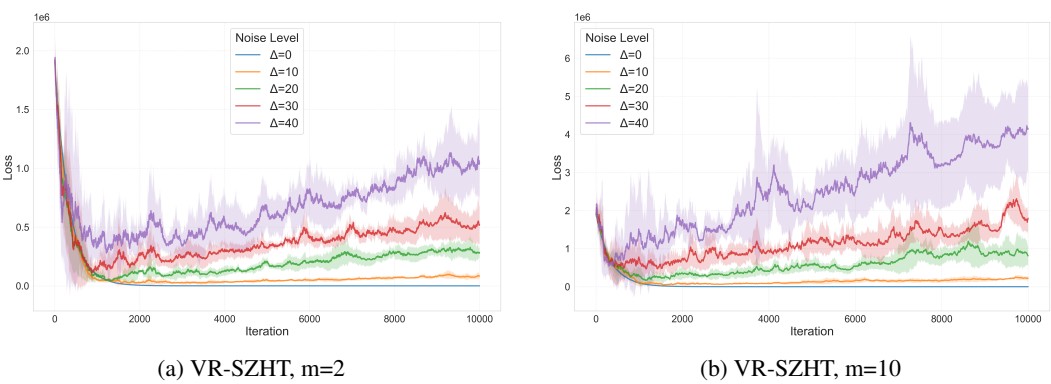

(a) VR-SZHT, m=2          (b) VR-SZHT, m=10

Figure 7: Set $\eta = 10^{-6}, n = 100, d = 100, k = 30, q = 20, \mu = 10^{-4}$.

Additionally, we extend experiment to high-dimensional and large-data scenarios.

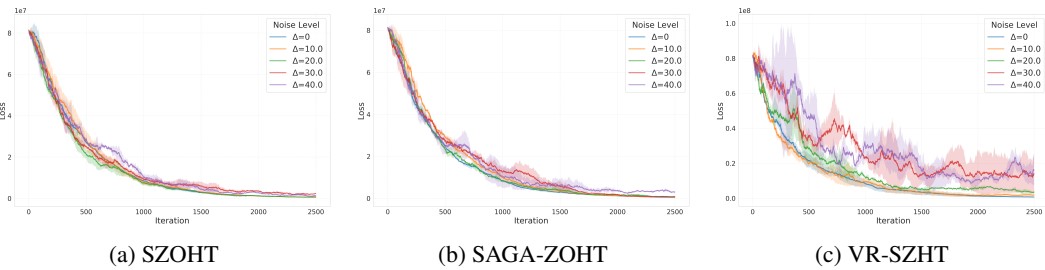

(a) SZOHT  (b) SAGA-ZOHT  (c) VR-SZHT

Figure 8: Set $\eta = 10^{-7}, n = 1,000,000, d = 10,000, k = 200, q = 20, \mu = 10^{-4}, m = 5$.

### E.2 FEW PIXELS ADVERSARIAL ATTACKS AGAINST DEFENSE

In few pixels adversarial attacks against defense experiment, the attack success rate over 100 images is shown in table 1. We also provide some example perturbed images in figure 10.

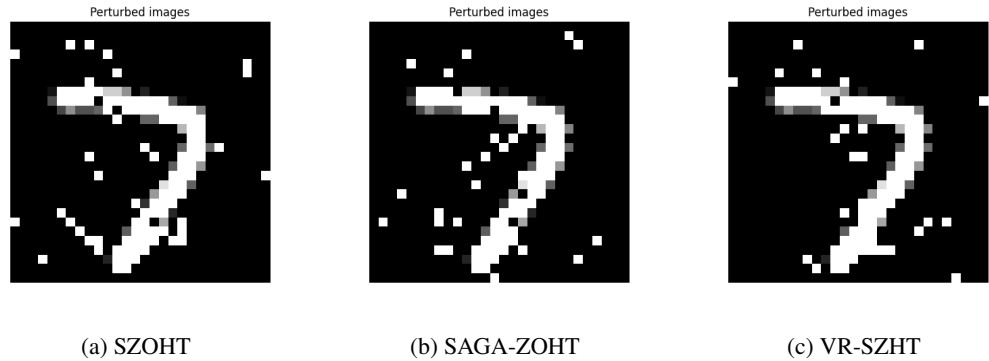

(a) SZOHT  (b) SAGA-ZOHT  (c) VR-SZHT

Figure 9: (a) is determined as number 3, while (b)(c) are still determined as number 7.

Table 1: Attack success rate under different $\nu$.

| $\nu$ | ALGORITHMS | ATK SUCCESS RATE |
|---|---|---|
| | SZOHT | 100/100 |
| 0 | SAGA-ZOHT | 99/100 |
| | VR-SZHT | 98/100 |
| | SZOHT | 87/100 |
| 0.1 | SAGA-ZOHT | 64/100 |
| | VR-SZHT | 74/100 |
| | SZOHT | 75/100 |
| 0.2 | SAGA-ZOHT | 49/100 |
| | VR-SZHT | 58/100 |

Additionally, we extend experiment to Cifar10. We demonstrated the loss function for adversarial attacks under undefended, poorly defended, and heavily defended conditions (the larger the better).

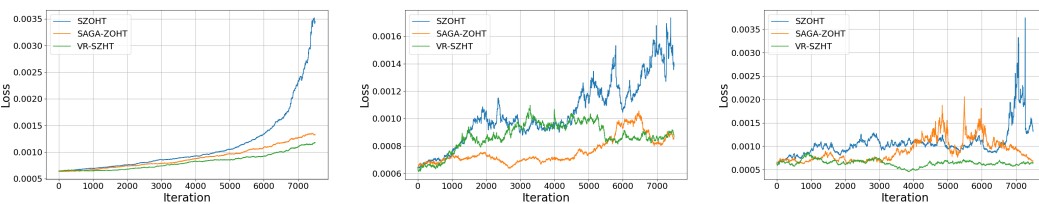

Figure 10: Adversarial attacks among SZOHT, SAGA-ZOHT, and VR-SZHT.

## F    LARGE LANGUAGE MODEL USAGE STATEMENT

In this work, we employed LLM to revise and refine text composition.

