# OpenReview forum: "Why Variance Reduction Hurts Noisy Zeroth-Order Hard-Thresholding?"
_ICLR.cc/2026/Conference — Submitted to ICLR 2026_

### Official Review · Reviewer_trYR · 2025-10-31

**Soundness:** 3
**Presentation:** 3
**Contribution:** 2
**Rating:** 6
**Confidence:** 2

**Summary:**

This work considers the problem of zeroth-order hard-theshlding in presence of noise, that is when 1) approximating gradients by computing finite differences of a random subset of components, 2) setting parameters to zero when they are small in magnitude, to satisfy a L0 penalty on the loss, and 3) The loss is stochastic, perhaps because of data batching.
In particular, the paper studies theoretically convergence rates of known algorithms (SZOHT), including variance reduction (VR-SZHT), and provides some empirical evaluations.
It shows that increasing the noise requires increasing the number of components used to estimate the gradient, and that, while variance reduction improves convergence, it also increases the algorithm’s sensitivity to noise.

**Strengths:**

- Paper is well written and clear

- Results appear to be correct and novel

- Theorem 2 is interesting. It shows that, while variance reduction improves convergence, it fails if the noise is too large, and the amount of variance reduction needs to be carefully adjusted to avoid this failure.

**Weaknesses:**

- Theorem 1 is quite unsurprising. It’s kind of obvious that increasing the noise requires increasing the number of components used to estimate the gradient (that effectively reduces the noise). However I am not able to comment on the technical contribution.

- There is no explanation on how to interpret experimental results. I didn’t quite understand them. Is the point that variance reduction VR-SZHT is worse than SZOHT (which is supposed to be the baseline)?

Minor:

- Broken reference on line 249

**Questions:**

NA

---

> ### Author Response · Authors · 2025-11-24
>
> Dear Reviewer trYR, we extend our sincere gratitude for dedicating your valuable time to reviewing our paper and for expressing your appreciation and support for our work. In the following, we will provide a comprehensive response to your review comments.
>
> > **Your comment:** Theorem 1 is quite unsurprising. It’s kind of obvious that increasing the noise requires increasing the number of components used to estimate the gradient (that effectively reduces the noise). However I am not able to comment on the technical contribution.
>
> **Our response:** Thank you for this comment. Theorem 1 presents the convergence results of the SZOHT algorithm in a noisy condition, which not only provides an intuitive conclusion that increasing the noise → increasing the required queries $q$, but also includes quantitative derivation and clear formulaic description. By introducing specific parameters $\lambda$ and $\rho$, it provides theoretical convergence limits and considers different system errors, accurately describing the specific relationship between noise magnitude and the number of gradient estimation directions. This analysis reveals the impact of noise on algorithm performance and clarifies how to adjust the queries $q$ to ensure the robustness of the algorithm in environments with high noise. Although $q$ of the SZOHT algorithm needs to increase with the increase of noise, it can still effectively handle noise and maintain stable convergence, which is of great significance in both theoretical and practical applications.
>
> However, beyond this, **a more significant contribution** of this work lies in the following insight: contrary to the prevailing consensus that variance reduction techniques (such as SVRG and SAGA) generally enhance stochastic optimization, our theoretical results (Theorems 2,3 and Remark 5) demonstrate that variance-reduced zeroth-order hard-thresholding algorithms (e.g., VR-SZHT and $\alpha$-$p$M-SZHT) actually tighten the upper bound on tolerable noise compared to the SZOHT method. This is because the reliance on historical gradient information in variance reduction amplifies the impact of noise, and this detrimental effect outweighs the benefits brought by variance reduction.
>
> > **Your comment:** There is no explanation on how to interpret experimental results. I didn’t quite understand them. Is the point that variance reduction VR-SZHT is worse than SZOHT (which is supposed to be the baseline)?
>
> **Our response:** Our experimental results demonstrate that the variance-reduced zeroth-order hard-thresholding algorithm **VR-SZHT performs comparably to or slightly better than the baseline SZOHT under low-noise or even noiseless conditions**. However, its performance may be **inferior to that of SZOHT in high-noise settings**. This observation aligns with our theoretical conclusion: while variance reduction enhances convergence, it introduces a trade-off by increasing the algorithm's sensitivity to noise, thereby compromising its robustness.
>
> As discussed in Section 4.3, the historical correction term in variance-reduced algorithms dominates under low noise levels, leading to more stable gradient estimates. In contrast, under significant noise, the influence of noise introduced by the historical term becomes amplified. This issue is further exacerbated by the hard-thresholding operator's inherent sensitivity to gradient variance, ultimately resulting in inferior convergence quality compared to SZOHT. This also confirms the greatest contribution of this article: variance reduction hurts noisy zeroth-order hard-thresholding.
>
> > **Your comment:** Broken reference on line 249
>
> **Our response:** Thank you for pointing out the broken reference on line 249. We apologize for the oversight. We have removed incorrect references and revised the wording (changes in blue).

---

> ### Author Response · Authors · 2025-11-27
>
> Dear Reviewer, this is a polite follow-up on the submitted rebuttal. We look forward to any further comments you may have.

---

### Official Review · Reviewer_w9SD · 2025-11-02

**Soundness:** 3
**Presentation:** 3
**Contribution:** 2
**Rating:** 4
**Confidence:** 4

**Summary:**

This submission studies the behavior of stochastic and variance-reduced zeroth-order hard-thresholding (ZOHT) algorithms under noisy conditions. The authors extend the stochastic ZOHT (SZOHT) framework to handle noise, providing convergence and complexity analysis under standard assumptions of restricted strong convexity and smoothness. They then generalize to variance-reduction variants—VR-SZHT (SVRG-style) and pM-SZHT (q-memorization-style)—and theoretically show that, contrary to intuition, variance reduction actually lowers the tolerable noise upper bound, thereby worsening performance in noisy environments. Experiments on synthetic regression, black-box adversarial attacks, and biological gene expression data confirm that simpler SZOHT is more robust than its variance-reduced counterparts.

**Strengths:**

Three strong points are as follows:

1. The paper demonstrates both analytically and empirically that variance-reduction techniques, though typically beneficial, can be harmful in noisy zeroth-order sparse optimization, offering a new understanding of the interplay between noise, gradient estimation, and sparsity.

2. The authors provide complete convergence proofs for SZOHT and its variants, establishing explicit bounds on the required number of random directions and the tolerable noise level.

3. Experimental results across three different tasks (synthetic regression, adversarial attacks, and noisy gene expression) consistently support the theoretical findings.

**Weaknesses:**

Three weak points are as follows:

1. The empirical section, though diverse in task type, remains small-scale; the algorithms are not tested on high-dimensional or large-data scenarios where zeroth-order methods are especially appealing.

2. The convergence results rely on restricted strong convexity/smoothness and bounded additive noise assumptions, which may not hold in highly nonconvex or structured real-world problems (the gene expression dataset used in this case).

3. The applicability of the proposed algorithms is very limited in the modern ML community. I was wondering if the authors could elaborate it on this.

**Questions:**

Q1. The discussion in the conclusion, suggesting that the results may inspire progress in “zeroth-order fine-tuning of Large Language Models,” seems overstated. The connection between the current theoretical setting (sparse recovery under additive noise with restricted strong convexity) and practical LLM fine-tuning is unclear. Could the authors discuss this in more detail?

Please see others in the weak points.

---

> ### Author Response · Authors · 2025-11-24
>
> Dear Reviewer w9SD, thank you for your insightful review. We sincerely hope that the following response can clarify the main concerns raised during the review.
>
> > **Your comment:** The empirical section, though diverse in task type, remains small-scale; the algorithms are not tested on high-dimensional or large-data scenarios where zeroth-order methods are especially appealing.
>
> **Our response:** Thank you for this insightful and professional comment. We fully agree that zeroth-order methods are particularly relevant in **high-dimensional** or **large-data** scenarios. To better reflect this aspect, we have substantially extended our empirical study. Concretely, we increased the scale of the sensitivity analysis experiment **from a data size of 100 to 1,000,000** and **from a dimension of 100 to 10,000**. In addition, we extended the pixels adversarial attack experiment **from MNIST (dimension = 784) to CIFAR-10 (dimension = 3072)**, which provides a more challenging high-dimensional setting. The updated experimental results and discussion have been shown in the anonymous link https://anonymous.4open.science/r/Additional-experimental-results-32EF and added to the appendix in the revised version of the paper (in blue).  The results of extended experiments shows, these zeroth-order hard-thresholding methods still effective in high-dimensional and large-data scenarios. And the results in this scenarios are also consistent with our theory that variance reduction worse performance in high noisy environments. As the dimensionality grows, these methods maintain efficiency by relying solely on function evaluations, avoiding issues like gradient instability or slow convergence. Our extended experiments on larger datasets, further demonstrate these zeroth-order methods can effectively handle complex, high-dimensional problems, offering a scalable solution for real-world applications.

---

> ### Author Response · Authors · 2025-11-24
>
> > **Your comment:** The convergence results rely on restricted strong convexity/smoothness and bounded additive noise assumptions, which may not hold in highly nonconvex or structured real-world problems (the gene expression dataset used in this case).
>
> **Our response:** We understand your concerns. However, the **restricted strong convexity/smoothness (RSC/RSS)** conditions are **relaxed versions** of the standard strong convexity and strong smoothness conditions. They only need to hold in **small, low-dimensional subspaces $s=2k+k^\ast$** rather than globally across all dimensions $d$. They are also common assumptions in high-dimensional statistics/sparse learning [1,2], widely used and supported in estimation in sparse linear regression, group sparse regression, matrix completeness, and generalized linear models [3].
>
> For example, in **recovery of sparse signal problems**, we usually assume that the $k$-sparse signal $x$ and the measurement matrix $A$ satisfy the RIP (Restricted Isometry Property) condition: $(1-\delta_k)\|x\|^2_2 \leq \|Ax\|^2_2 \leq (1+\delta_k)\|x\|^2_2$, which approximates that the objective function in the optimization problem having **RSC** and **RSS** properties [4,5]. Bounded additive noise is a commonly assumed condition in optimization problems [6,7], where the observations or measurements of the signal are subject to perturbations within a specified range, thus naturally modeling the noise in a bounded form [8].
>
> As for **the gene expression dataset in our article**, the noise in gene expression data is introduced by manual detection, and the detection error of each gene expression data has a limited impact on the prediction model. Therefore, it can be assumed that the noise is bounded. In our setting, we evaluate the additive noise of  different method for filling missing data and calculated that $\Delta=2.56e-4$. Meanwhile, the gene expression matrix is naturally sparse, meaning that the expression data of most genes can be approximated by a few main components. Although the gene matrix is high-dimensional, the actual number of genes that affect diseases is relatively small. This means that the optimization problem has good geometric properties in low rank matrix space. To verify it, we calculate the sample derivatives and use them to construct a weighted design matrix. The Gauss-Newton Hessian is approximated using $H \approx \frac{1}{n} A'^T A'$, where $A'$ is the weighted matrix derived from the sample gradients. For different sparsity levels $s$, we randomly sample sparse subsets of features, compute the corresponding submatrices, and calculate the Gram matrix. We then estimate the minimum and maximum eigenvalues of these submatrices, which provide the **RSC** and **RSS** constants $m_s$ and $L_s$. The results are tabulated for various sparsity levels, and **the assumptions are considered valid if both constants $m_s,L_s$ are positive and bounded**.
>
>
> | Sparsity ($s$) | $ m_s $             | $ L_s $             |
> |--------------|-----------------------|-----------------------|
> | 2            | 0.15              | 0.51              |
> | 10           | 0.06              | 0.44              |
> | 20           | 0.03              | 0.64              |
> | 30           | 0.03              | 0.69              |
> | 40           | 0.02             | 1.26              |
>
> The experimental results demonstrate that the **RSC/RSS** assumptions holds true at **higher sparsity levels**.

---

> ### Author Response · Authors · 2025-11-24
>
> > **Your comment:** The applicability of the proposed algorithms is very limited in the modern ML community. I was wondering if the authors could elaborate it on this.
>
> **Our response:**
>
> Hard-thresholding methods have been proven to be efficient and practical in tasks such as **low-rank matrix recovery and compressed sensing** [1,9]. Zeroth-order methods, which offer unique advantages when gradients are unavailable or impractical to compute, are particularly relevant in real-world tasks such as **black-box adversarial attacks, LLM fine-tuning, and reinforcement learning** [10]. Real-world tasks also often involve various sources of noise that can affect algorithm performance. These sources include noisy data, label errors, and model hallucinations. Noisy data can cause the model to learn incorrect patterns, while label errors can mislead supervised learning tasks. In LLM scenarios, hard-thresholding can be applied to adaptation layers, gating mechanisms, or mask parameters, where only a small number of channels or blocks are updated in a black-box fashion. The zeroth-order hard-thresholding method directly operates on this sparsified parameterization, making it a feasible approach, especially when the target model is a black-box that does not support backpropagation.
>
> Previous research [11] has applied hard-thresholding methods to low-rank fine-tuning of LLMs, using hard thresholding after each gradient step without the need for additional memory and computation time. In addition, other study [12] have combined low-rank approximation with zeroth-order optimization techniques to enable large-scale model fine-tuning in settings where gradient information is either unavailable or expensive to compute. Additionally, variance reduction techniques have been introduced into zeroth-order frameworks to improve the fine-tuning of LLMs [13].
>
> To demonstrating the applicability of the proposed algorithms, we apply it to **zeroth-order hard-thresholding LoRA fine-tuning for black-box prompt optimization**. In this setup, the process begins by using a white-box model to generate a set of prompts, which are then fed into a black-box model to produce images. The generated prompts are input into the black-box model to produce images, and we evaluate the results by multiple indicators. Zeroth-order gradients are then calculated based on these results and used to perform LoRA fine-tuning on the white-box model. The low-rank matrices $A$ and $B$ are updated using zeroth-order gradients and applied hard-thresholding operation. In scenarios where multiple LoRA adapters are required, applying hard-threshold sparsity to LoRA adapters is particularly useful for optimizing memory storage. Specifically, we use Stable Diffusion v1.4 as the black-box model to generate images, while Llama-2-7b-hf, an open-source LLM, is employed to generate prompts. The results are shown as follows:
>
>
> | method       | Domain   | Aesthetic $\uparrow$ | PickScore $\uparrow$ | CLIP Score $\uparrow$ |
> |--------------|----------|-----------|-----------|------------|
> | Original Prompt           | Painting | 5.652     | 19.395    | 0.248      |
> | Zeroth-order LoRA + Hard-threshold  | Painting | 5.944     | 19.571    | 0.271      |
> | Original Prompt           | Art      | 5.453     | 19.381    | 0.237      |
> | Zeroth-order LoRA + Hard-threshold  | Art      | 5.887     | 19.485    | 0.253      |
> | Original Prompt           | Anime    | 5.433     | 19.391    | 0.236      |
> | Zeroth-order LoRA + Hard-threshold  | Anime    | 5.897     | 19.561    | 0.265      |

---

> ### Author Response · Authors · 2025-11-24
>
> > **Your comment:** The discussion in the conclusion, suggesting that the results may inspire progress in “zeroth-order fine-tuning of Large Language Models,” seems overstated. The connection between the current theoretical setting (sparse recovery under additive noise with restricted strong convexity) and practical LLM fine-tuning is unclear. Could the authors discuss this in more detail?
>
> **Our response:** Thank you for highlighting this important point. As discussed in the previous section, the current theoretical framework can be interpreted as a **sparse optimization LoRA adapter**. This connection implies that certain theoretical aspects of sparse recovery can be applied to fine-tuning procedures that update only a sparse subset of parameters. In many fine-tuning setups for large models, only a small subset of parameters is updated or **updates are restricted to a low-dimensional subspace** (e.g., sparse or low-rank updates). Methods such as LoRA, Adapter techniques, Prefix/Prompt Tuning, and BitFit fall into this category. This structural constraint is aligned with our theoretical setting of sparse recovery in a constrained, low-dimensional subspace.
>
> For instance, [14] utilized **sparse masking for the projection $B$** in the LoRA adapter, only a small fraction of parameters in matrices $B$ are updated, effectively integrating sparse settings with fine-tuning objectives in LLMs.
>
> Moreover, various sources of randomness and uncertainty (data noise, optimization noise, black-box query noise) are unavoidable in practical fine-tuning pipelines, and modeling them as bounded additive noise is a standard abstraction that enables convergence and error analyses. Consequently, zeroth-order hard-thresholding algorithms align naturally with several real-world fine-tuning constraints in LLMs, such as sparsity and limited memory resources.
>
> [1] Nguyen N, Needell D, Woolf T. Linear convergence of stochastic iterative greedy algorithms with sparse constraints. IEEE Transactions on Information Theory. 2017 Sep 5;63(11):6869-95.
>
> [2] Jie Shen and Ping Li. A tight bound of hardthresholding. Journal of Machine Learning Research, 18(208):1–42,2018.
>
> [3] Alekh Agarwal, Sahand Negahban, and Martin J. Wainwright. Fast global convergence of gradient methods for high-dimensional statistical recovery. The Annals of Statistics, 40(5):2452–2482, 2012.
>
> [4] Zhou S, Xiu N, Qi HD. Global and quadratic convergence of Newton hard-thresholding pursuit. Journal of Machine Learning Research. 2021;22(12):1-45.
>
> [5] Bahmani, S., Raj, B., and Boufounos, P. Greedy sparsity-constrained optimization. Journal of Machine Learning Research, 14:807–841, 2013.
>
> [6] Alexandre Belloni, Tengyuan Liang, Hariharan Narayanan, and Alexander Rakhlin. Escaping the local minima via simulated annealing: Optimization of approximately convex functions. In Conference on Learning Theory, pages 240–265. PMLR, 2015.
>
> [7] Yaron Singer and Jan Vondrák. Information-theoretic lower bounds for convex optimization with erroneous oracles. In Advances in Neural Information Processing Systems, pages 3204–3212, 2015.
>
> [8] Cai TT, Xu G, Zhang J. On Recovery of Sparse Signals Via $\ell _ {1} $ Minimization. IEEE Transactions on Information Theory. 2009 Jun 16;55(7):3388-97.
>
> [9] Yuan XT, Li P, Zhang T. Gradient hard thresholding pursuit. Journal of Machine Learning Research. 2018;18(166):1-43.
>
> [10] Malladi S, Gao T, Nichani E, Damian A, Lee JD, Chen D, Arora S. Fine-tuning language models with just forward passes. Advances in Neural Information Processing Systems. 2023 Dec 15;36:53038-75.
>
> [11] Veprikov A, Solodkin V, Zyl A, Savchenko A, Beznosikov A. WeightLoRA: Keep Only Necessary Adapters. arXiv preprint arXiv:2506.02724. 2025 Jun 3.
>
> [12] Chen Y, Zhang Y, Cao L, Yuan K, Wen Z. Enhancing zeroth-order fine-tuning for language models with low-rank structures. arXiv preprint arXiv:2410.07698. 2024 Oct 10.
>
> [13] Gautam T, Park Y, Zhou H, Raman P, Ha W. Variance-reduced zeroth-order methods for fine-tuning language models. arXiv preprint arXiv:2404.08080. 2024 Apr 11.
>
> [14] Zhang J, You J, Panda A, Goldstein T. Lori: Reducing cross-task interference in multi-task low-rank adaptation. arXiv preprint arXiv:2504.07448. 2025 Apr 10.

---

> ### Author Response · Authors · 2025-11-27
>
> Dear Reviewer, this is a polite follow-up on the submitted rebuttal. We look forward to any further comments you may have.

---

### Author Response · Authors · 2025-11-30
**Response to Area Chair**

Dear Area Chair,
We acknowledge the additional workload that the recent ICLR review process has placed upon you and sincerely appreciate the immense responsibility and challenges you are undertaking. Your efforts are crucial in maintaining the fairness and integrity of the machine learning community, and we are truly grateful for your dedication. To facilitate your evaluation of our paper, we have summaried the **concerns raised by reviewers** and **our responses** as following:

1. **Reasonableness of restricted strong convexity/smooth (RSC/RSS) and Bounded Additive Noise Assumption in real-world problems:**(w9SD)
    - We clarify that these **assumptions are common** in many machine learning algorithms and reflects preactical settings.
    - We conducted real-world problem experiment to **verify the assumptions**.
2. **Algorithm's Applicability to Modern Tasks:**(w9SD)
    - We have demonstrated that the zeroth-order hard-thresholding algorithms has practical applications in many **black-box sparsity scenarios**.
    - We have included new **black-box sparse LoRA optimization experiments** to demonstrate the practicality of the algorithms.
3. **Limited Scale of Experiments and Data:**(w9SD)
    - We have significantly expanded our experiments to include high-dimensional and large-scale datasets.
4. **Connection Between Theoretical Setup and Practical LLM Fine-Tuning:**(w9SD)
    - We provide a clearer connection between our theoretical analysis and practical applications by demonstrating how our theoretical framework applies to actual fine-tuning scenarios for LLM.
5. **Difficulty in Discerning the Technical Contribution:**(trYR)
    - We have clarified our contributions, emphasizing the extension of zeroth-order hard-thresholding algorithms to noisy settings and the **new convergence analysis** we provide.
    - We have discovered the **astonishing fact** that variance reduction not only does not necessarily accelerate convergence, but may also reduce the upper bound of tolerable noise.
6. **Difficulty in Interpreting Experimental Outcomes:**(trYR)
    - We provide a detailed interpretation of the experimental results, particularly focusing on the key insight that **variance reduction techniques like VR-SZHT reduce noise tolerance**. This is the core reason behind the performance differences observed in our experiments.

***

**Key Theoretical Insight and Experimental Clarifications**
- Our key theoretical insight is that variance reduction methods reduce the upper bound of noise tolerance, contrary to the expectation that they should improve performance in noisy settings. This insight directly explains why algorithms like VR-SZHT perform worse than SZOHT under noisy conditions.

**Conclusion and Request for Reevaluation**

- We hope that these clarifications and additions address the reviewers' concerns and provide a clearer understanding of our work's contributions. We believe that our paper offers valuable insights into the behavior of stochastic and variance reduction zeroth-order hard-thresholding algorithms in noisy environments and provides a solid theoretical foundation for future work in this area.

We kindly request that you reevaluate our submission in light of these revisions. Thank you once again for your time and hard work. We truly hope that our work will contribute meaningfully to the machine learning community.

Best regards,

The Authors

---

### Meta-Review · Area_Chair_NxW5 · 2026-01-06

**Summary:**

The paper analyzes stochastic and variance-reduced zeroth-order hard-thresholding methods in noisy settings. It extends SZOHT with convergence and complexity guarantees under standard restricted strong convexity and smoothness assumptions, and introduces variance-reduced variants. The main theoretical finding is that variance reduction counterintuitively reduces the admissible noise level, degrading robustness. Empirical results across synthetic regression, black-box attacks, and gene expression data corroborate that plain SZOHT outperforms its variance-reduced versions under noise.

**Reviewer Concerns:**

**Reviewer w9SD.** The reviewer questioned the small-scale nature of the experiments, the reliance on restrictive theoretical assumptions (RSC/RSS and bounded noise), and the arguably overstated relevance to modern ML and LLM fine-tuning. The authors addressed the first two concerns by adding higher-dimensional experiments and carefully justifying the assumptions through sparse recovery theory and empirical checks on real data. They also clarified the LLM connection by framing their method as sparse/low-rank adaptation (e.g., LoRA) and adding a zeroth-order fine-tuning example, though the practical relevance to large-scale LLM training remains somewhat indirect.

**Reviewer trYR.** The reviewer questioned the novelty of Theorem 1, finding it unsurprising, and found the experimental results unclear, particularly regarding whether VR-SZHT is worse than the baseline SZOHT; a minor broken reference was also noted. The authors responded by clarifying that Theorem 1 provides a quantitative convergence analysis under noise and by emphasizing their main contribution: showing that variance reduction can harm zeroth-order hard-thresholding in high-noise regimes. They also clearly explained that VR-SZHT matches or slightly outperforms SZOHT at low noise but underperforms at high noise, and they fixed the reference, though concerns about the incremental nature of Theorem 1 may remain.

**Reviewer Scores:**

Based on my understanding, the concerns of the reviewers were resolved only partially. Therefore, I think, they would keep the original score. Therefore, I recommend rejection of this paper.

---

### Decision · Program_Chairs · 2026-01-26

Reject